# A protease and a lipoprotein jointly modulate the conserved ExoR-ExoS-ChvI signaling pathway critical in *Sinorhizobium meliloti* for symbiosis with legume hosts

Julian A. Bustamante[1][☯], Josue S. Ceron[1][☯], Ivan Thomas Gao[1][☯], Hector A. Ramirez[1][☯], Milo V. Aviles[1], Demsin Bet Adam[1], Jason R. Brice[1], Rodrigo A. Cuellar[1], Eva Dockery[1], Miguel Karlo Jabagat[1], Donna Grace Karp[1], Joseph Kin-On Lau[1], Suling Li[1], Raymondo Lopez-Magaña[1], Rebecca R. Moore[1], Bethany Kristi R. Morin[1], Juliana Nzongo[1], Yasha Rezaeihaghighi[1], Joseph Sapienza-Martinez[1], Tuyet Thi Kim Tran[1], Zhenzhong Huang[1], Aaron J. Duthoy[2], Melanie J. Barnett[2], Sharon R. Long[2], Joseph C. Chen[1]*

1 Department of Biology, San Francisco State University, San Francisco, California, United States of America, 2 Department of Biology, Stanford University, Stanford, California, United States of America

☯ These authors contributed equally to this work.
* chenj@sfsu.edu

**Data Availability Statement:** Microarray data have been deposited in the Gene Expression Omnibus (GEO) under accession GSE155833. All plasmid

## Abstract

*Sinorhizobium meliloti* is a model alpha-proteobacterium for investigating microbe-host interactions, in particular nitrogen-fixing rhizobium-legume symbioses. Successful infection requires complex coordination between compatible host and endosymbiont, including bacterial production of succinoglycan, also known as exopolysaccharide-I (EPS-I). In *S. meliloti* EPS-I production is controlled by the conserved ExoS-ChvI two-component system. Periplasmic ExoR associates with the ExoS histidine kinase and negatively regulates ChvI-dependent expression of *exo* genes, necessary for EPS-I synthesis. We show that two extracytoplasmic proteins, LppA (a lipoprotein) and JspA (a lipoprotein and a metalloprotease), jointly influence EPS-I synthesis by modulating the ExoR-ExoS-ChvI pathway and expression of genes in the ChvI regulon. Deletions of *jspA* and *lppA* led to lower EPS-I production and competitive disadvantage during host colonization, for both *S. meliloti* with *Medicago sativa* and *S. medicae* with *M. truncatula*. Overexpression of *jspA* reduced steady-state levels of ExoR, suggesting that the JspA protease participates in ExoR degradation. This reduction in ExoR levels is dependent on LppA and can be replicated with ExoR, JspA, and LppA expressed exogenously in *Caulobacter crescentus* and *Escherichia coli*. Akin to signaling pathways that sense extracytoplasmic stress in other bacteria, JspA and LppA may monitor periplasmic conditions during interaction with the plant host to adjust accordingly expression of genes that contribute to efficient symbiosis. The molecular mechanisms underlying host colonization in our model system may have parallels in related alpha-proteobacteria.

sequences are provided as supporting information (S1 File).

**Funding:** Research reported in this publication was supported by the National Institute of General Medical Sciences of the National Institutes of Health (NIH) under Award Number SC3-GM096943 to J.C.C. and National Science Foundation (NSF), Division of Integrative Organismal Systems, under Rules of Life Award Number 2015870 to S.R.L. NIH Award Number T34-GM008574 (MARC) provided support for J.A. B., J.S.C., R.A.C., R.L.-M., and J.N.; R25-GM048972 (Bridge to the Doctorate) supported H. A.R.; R25-GM059298 (MBRS-RISE) supported J. S.-M.; R25-GM050078 (Bridges to the Baccalaureate) supported J.S.C. and R.A.C.; T34-GM145400 (U-RISE) supported R.A.C.; and UL1-GM118985, TL4-GM118986, RL5-GM118984 (SF BUILD) supported B.K.R.M. NSF REU DBI Award Number 1156452 provided summer funding for R. L.-M. and R.B., and NSF DBI Award Number 1548297 provided summer funding for H.A.R. R.R. M. was supported by the Beckman Scholars Program. The content is solely the responsibility of the authors and does not necessarily represent the official views of the funding agencies. The funders had no role in study design, data collection and analysis, decision to publish, or preparation of the manuscript.

**Competing interests:** The authors have declared that no competing interests exist.

## Author summary

Symbiotic bacteria that live in the roots of legume plants produce biologically accessible nitrogen compounds, offering a more sustainable and environmentally sound alternative to industrial fertilizers generated from fossil fuels. Understanding the multitude of factors that contribute to successful interaction between such bacteria and their plant hosts can help refine strategies for improving agricultural output. In addition, because disease-causing microbes share many genes with these beneficial bacteria, unraveling the cellular mechanisms that facilitate host invasion can reveal ways to prevent and treat infectious diseases. In this report we show that two genes in the model bacterium *Sinorhizobium meliloti* contribute to effective symbiosis by helping the cells adapt to living in host plants. This finding furthers knowledge about genetic factors that regulate interactions between microbes and their hosts.

## Introduction

Rhizobia-legume symbioses account for a substantial proportion of terrestrial nitrogen fixation, reducing molecular dinitrogen to a more bioavailable form such as ammonia [1,2]. The mutualistic relationship requires complex communication and coordination between two compatible partners [3,4], as well as bacterial adaptation to the "stresses" of the host plant environment [5,6]. The alpha-proteobacterium *Sinorhizobium meliloti* and its hosts, including *Medicago sativa* (alfalfa) and *M. truncatula* (barrel medic), emerged as models for nitrogen-fixing root nodule symbiosis [7]. Here, compounds released by the host plant induce bacterial production of signaling molecules called Nod factors, required for eliciting formation of root nodules [8]. Nodule colonization begins with bacterial cells invading the root hair via plant cell wall-derived tunnels called infection threads, followed by release into plant cells, in which the rhizobia differentiate into "bacteroids" capable of fixing nitrogen in exchange for carbon from the host [2,9,10].

Multiple factors found to be critical for *S. meliloti* to form mutualistic symbiosis have been shown to contribute to host infection in related pathogens, such as *Brucella* spp., suggesting mechanistic parallels between mutualism and pathogenesis [7]. One such shared mechanism is the ExoS-ChvI two-component phosphorelay pathway, conserved across many alpha-proteobacteria, particularly in *Rhizobiales* (synonym *Hyphomicrobiales*) (Fig 1) [11,12]. ExoS is a membrane-bound histidine kinase with a periplasmic sensor domain, while ChvI is its cognate response regulator [13]. Mutations in ExoS and ChvI, as well as their orthologs in related endosymbionts, impair host colonization [14–22]. A third component of the *S. meliloti* signaling system, ExoR, acts as a periplasmic repressor of ExoS via physical association [19,21]. ExoR is regulated by proteolysis [23–25], and binding to ExoS protects it from degradation [19]. Mutations in ExoR also disrupt symbiosis [19,21,26–28].

Cues that suggest transition into the host environment appear to stimulate the ExoR-ExoS-ChvI signaling cascade to promote a developmental shift from free-living to symbiotic [12]. However, conditions that specifically trigger the ExoR-ExoS-ChvI pathway in *S. meliloti* remain elusive [17,29,30]. Furthermore, different cues for divergent species are possible, and some cues may directly activate the ExoS sensor kinase and bypass ExoR [31–35].

Irrespective of the specific triggers, the *S. meliloti* ExoR-ExoS-ChvI system influences a multitude of physiological activities, including exopolysaccharide (EPS) production, motility, biofilm formation, cell envelope maintenance, and nutrient utilization, befitting its pivotal regulation of symbiotic development [17,18,21,26]. Initial transcriptome profiles of *S. meliloti*

**Fig 1. Model of how JspA and LppA influence the ExoR-ExoS-ChvI signaling pathway.** Schematic diagram shows relationship of pathway components, their subcellular locations, and impact on expression of representative genes. Pointed and blunt arrowheads represent positive and negative regulation, respectively. Solid arrows indicate previously demonstrated, direct interactions. Results from this study suggest that, in response to cell envelope stress such as exposure to acidic pH, JspA and LppA negatively regulate ExoR via proteolysis. As a typical pair of histidine kinase and response regulator, ExoS and ChvI are presumed to function as homodimers [13,30]; for simplicity, the diagram does not show that.

*exoS*::Tn5 and *exoR*::Tn5 mutants revealed altered expression of hundreds of genes [21,26], but subsequent interrogation that included identification of genomic regions bound by ChvI winnowed the direct targets of the response regulator down to 64, many known to participate in physiological activities described above [30,36]. Perhaps illustrating the complex interaction of regulatory pathways and the difficulty of signal deconvolution, a significant fraction of ChvI targets also changed expression with other published perturbations [30], including acid stress [37–39], antimicrobial peptide treatment [40], phosphate starvation [41], cyclic nucleotide accumulation [42], overexpression of SyrA [43], and mutations in *podJ*, *cbrA*, *ntrY*, and *emrR* [44–47].

One key subset of the regulon induced upon ExoS-ChvI activation is the *exo* genes, responsible for synthesis of succinoglycan, or EPS-I, originally characterized in *S. meliloti* strain Rm1021 as the only symbiotically active EPS [48–51]. An increase in EPS-I production, usually concomitant with a decrease in flagellar motility [30,44], represents a physiological transition from saprophytic to endosymbiotic, as EPS-I contributes to successful interaction between compatible symbiotic partners. Mutants that lack EPS-I or synthesize variants with altered structures (for example, absence of succinylation) exhibit defects in the initiation or elongation of infection threads, while changes in EPS-I levels can influence symbiotic efficiency [52–56]. Thus, both the quality and quantity of EPS-I matter during infection. EPS-I may serve as a recognition signal, particularly for suppressing host defenses [57]. While no plant receptor for *S. meliloti* EPS-I has been identified so far [58], EPS-I does enhance tolerance of various environmental assaults [59,60], including those encountered during host colonization, such as acidity, oxidative stress, and antimicrobial peptides [61–65].

In particular, EPS-I confers resistance to the antimicrobial activity of NCR247 [63,65], which belongs to a diverse family of small, nodule-specific cysteine-rich (NCR) peptides encoded by certain legumes [10,66,67]. Structurally similar to host defensins [68], different NCR peptides regulate bacterial load in nodules and influence distinct aspects of terminal bacteroid differentiation, including maintaining survival and preventing premature senescence [69–74]. In addition to EPS-I, other bacterial factors can modulate the effects of NCR peptides [63,75–77].

One of the genes previously identified in a transposon-based screen as necessary for *S. meliloti* resistance against NCR247 in culture is SMc03872 (*jspA*), predicted to encode a periplasmic protease conserved in alpha-proteobacteria and shown to confer a competitive advantage during symbiosis with alfalfa [63]. *jspA* was also identified in a genetic selection for suppressors that ameliorated the osmosensitivity of a *podJ* null mutant [44]. That work demonstrated that PodJ is a conserved polarity factor that contributes to cell envelope integrity and EPS-I production in *S. meliloti*, and that deletion of *jspA* or SMc00067 (*lppA*), both encoding putative lipoproteins, reduced EPS-I levels. Here we show that *jspA* and *lppA* jointly influence EPS-I production by lowering the steady-state levels of periplasmic ExoR and thus activating the ExoS-ChvI signal transduction pathway (Fig 1). This regulation contributes to competitive fitness during host colonization, suggesting that *jspA* and *lppA* facilitate transition to a gene expression pattern more suitable for the host environment.

## Results

### LppA and JspA jointly contribute to EPS-I biosynthesis and symbiotic competitiveness

In a previous suppressor analysis to identify mutations that alleviated the cell envelope defects associated with the *podJ1* deletion, we found two genes (SMc00067 and SMc03872) whose interruption or deletion led to consistent and significant reduction in EPS-I production [44].

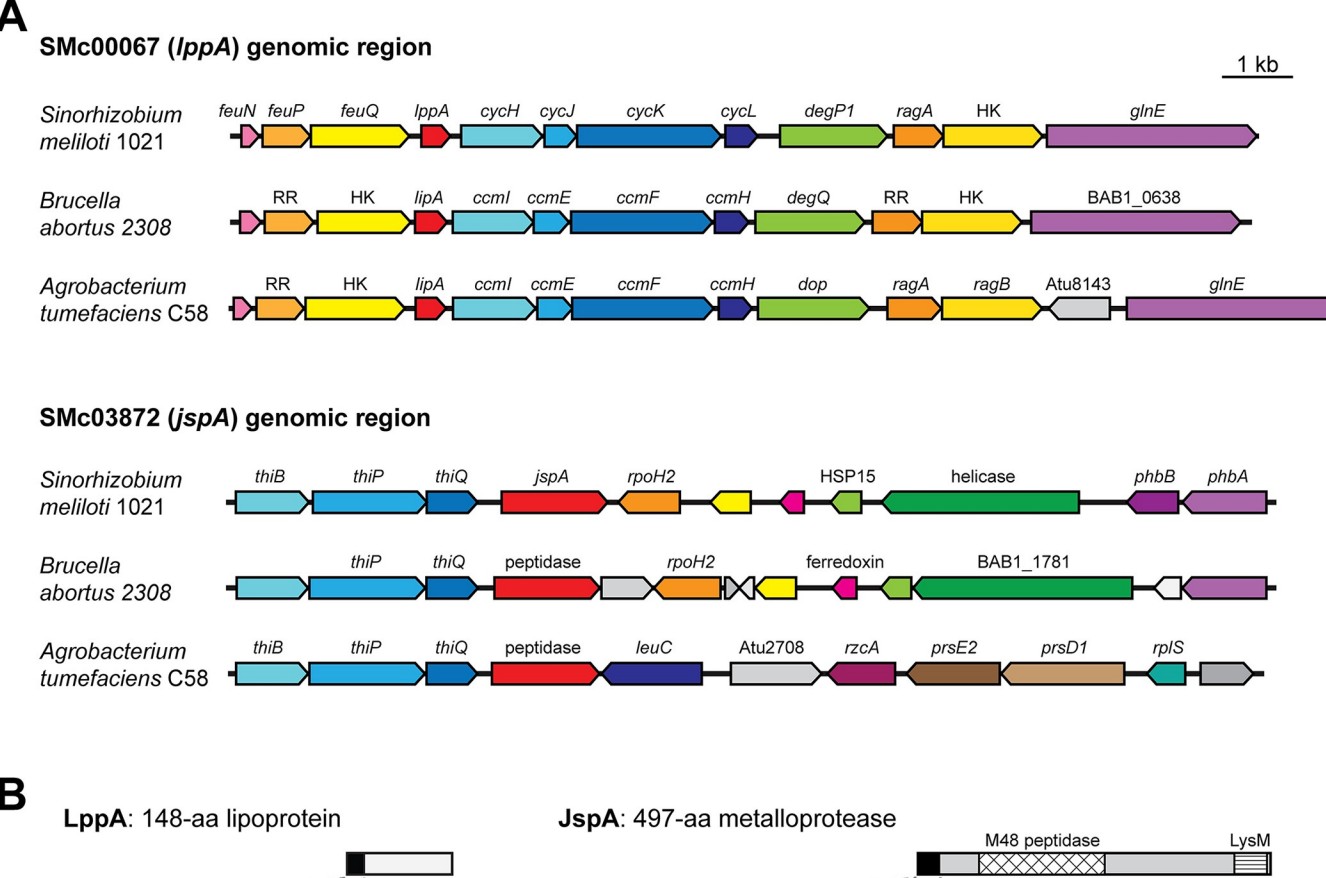

**Fig 2. Schematics of the genomic regions around *lppA* and *jspA* and of their protein products. (A)** *S. meilioti lppA* (SMc00067) and *jspA* (SMc03872) share synteny with their respective orthologs in closely related alpha-proteobacteria, such as *B. abortus* and *A. tumefaciens*. Gene and ORF names are shown as annotated, with pentagonal arrows indicating directionality. Arrows with the same colors in different species represent probable homologs, with red arrows indicating *lppA* or *jspA* orthologs; genes without annotated functions or obvious orthologs in corresponding regions are depicted with shades of grey. RR and HK signify response regulators and histidine kinases. The drawing is to scale; bar indicates 1 kb. **(B)** *lppA* encodes a 148-aa lipoprotein, while *jspA* encodes a 497-aa metalloprotease. Both LppA and JspA contain lipoprotein signal peptides at their N-termini; the sequences of these leader peptides are shown, with red arrows indicating cleavage sites before the invariant cysteine of the lipobox motifs, underlined. The N-terminus of LppA was originally annotated as the 13th amino acid ($V_{13}$) shown here, but extension of 12 amino acids provides a better signal sequence. JspA also contains M48 peptidase and LysM domains; key amino acids of the peptidase domain are displayed. Grey numbers indicate residues that border the predicted protein domains.

SMc00067 (annotated as *lppA*) encodes a 148-amino acid lipoprotein, while SMc03872 (here named *jspA*, for *podJ* *s*uppressor *p*rotease *A*) encodes a 497-amino acid metalloprotease that contains an M48 peptidase domain, with a conserved HEXXH active site, and a LysM domain, commonly associated with peptidoglycan binding (Fig 2) [63,78,79]. BLAST searches against representative bacterial species indicated that both genes are highly conserved within the *Rhizobiales* group of alpha-proteobacteria, based on shared synteny and protein sequences (Fig 2A and S1 Table) [80]. Outside of the *Rhizobiales* group, orthologs of LppA were rare or difficult to identify, while the sequence similarities of JspA homologs were generally lower than those found within *Rhizobiales* (S1 Table). Both LppA and JspA contain lipoprotein signal peptides at their N-termini, each with a stretch of hydrophobic amino acids followed by an invariant Cys within the lipobox motif (Fig 2B) [81]. In the original annotation for LppA in *S. meliloti* Rm1021, the protein starts seven codons upstream of the LAGC lipobox, with

VVASGVA, but N-terminal extension of 12 codons adds more hydrophobic amino acids, allowing a more optimal signal sequence; thus, we have numbered the amino acid sequence accordingly. The lipoprotein signals suggest that each protein is exported out of the cytoplasm and attached to the inner or outer membrane [82].

To verify that LppA and JspA contribute to EPS-I production, we performed complementation analysis by introducing plasmids carrying *lppA* or *jspA* under the control of a taurine-inducible promoter (P$_{tau}$) [83] into wild-type Rm1021 or deletion mutants. Serial dilutions of liquid cultures were spotted onto plates containing calcofluor, which fluoresces when bound to EPS-I [84]. Consistent with previously published results [44], Δ*lppA* or Δ*jspA* mutants carrying the empty vector exhibited lower levels of EPS-I production, with 60–70% of calcofluor fluorescence compared to wild type carrying the vector, in the presence or absence of the inducer (Fig 3A and 3B and S2 Table). Compared to their counterparts with the vector, wild-type and mutant strains with plasmids carrying *lppA* or *jspA* showed elevated fluorescence levels in the presence of taurine (Fig 3A and 3B); this increase in fluorescence was not observed in the absence of taurine (S2 Table). Similar results were obtained with the closely related species *S. medicae* strain WSM419: deletion of the *lppA* or *jspA* ortholog in that wild-type strain reduced fluorescence on plates containing calcofluor (S1A Fig), and complementation with the heterologous *S. meliloti* gene rescued the defect (S1B Fig, strains JOE5290 and JOE5264). At higher taurine concentrations (5 and 10 mM), induction of *jspA* expression in *S. meliloti* Rm1021 inhibited colony formation (Fig 3C), but we did not observe this effect with *lppA* expression, even with the highest possible concentration of taurine (100 mM) (Fig 3A). While induction of *lppA* or *jspA* expression promoted EPS-I production in the wild type or corresponding deletion mutants, *lppA* expression in the Δ*jspA* mutant (Fig 3A, bottom row) and *jspA* expression in the Δ*lppA* mutant (Fig 3B, bottom row) failed to increase calcofluor fluorescence (S2B Table). In addition, although overexpression of *jspA* at 10 mM taurine increased EPS-I production in the Δ*lppA* mutant, it did not cause growth arrest in that background (Fig 3C, bottom row). These results suggest that LppA and JspA require each other to stimulate EPS-I production and accomplish their physiological activities, albeit overexpression of JspA may be able to bypass some of the requirement for LppA.

Since EPS-I is critical for infection thread formation during host colonization, we asked if deletion of *lppA* or *jspA* leads to a symbiosis defect. *M. sativa* seedlings were inoculated with wild-type Rm1021 or Δ*lppA* or Δ*jspA* derivatives, and *M. truncatula* seedlings were inoculated with wild-type WSM419 or its Δ*lppA* or Δ*jspA* derivatives because WSM419 forms more efficient symbiosis with *M. truncatula* than *S. meliloti* strains [85–87]. For each host species, we did not observe obvious differences in plant growth and the development of root nodules on nitrogen-free medium over the course of four weeks: the average numbers of pink, nitrogen-fixing nodules per plant were similar 21 and 28 days after inoculation with wild-type or mutant strains (S3 Table), suggesting that the two genes are not required for symbiosis. To examine more closely if LppA and JspA contribute to efficient host colonization, we conducted competitive infection assays in which seedlings were inoculated with mixtures containing equal numbers of two strains, and bacteria were recovered from root nodules 28 days post-inoculation to determine occupancy rates (Materials and methods). One or both strains were marked with distinct antibiotic resistance to facilitate identification via plating after extraction from nodules. *M. sativa* plants were inoculated with mixtures of Rm1021 derivatives (Fig 4A), while *M. truncatula* plants were inoculated with mixtures of WSM419 derivatives (Fig 4B). Consistent with previous reports [44,88], we found that a small percentage of the nodules contained mixed populations of bacteria, while the majority of nodules were dominated by one of two strains. Discounting those nodules containing mixed populations, roughly equal numbers of nodules (45–55%) were occupied by each strain when the inoculum contained two wild-type

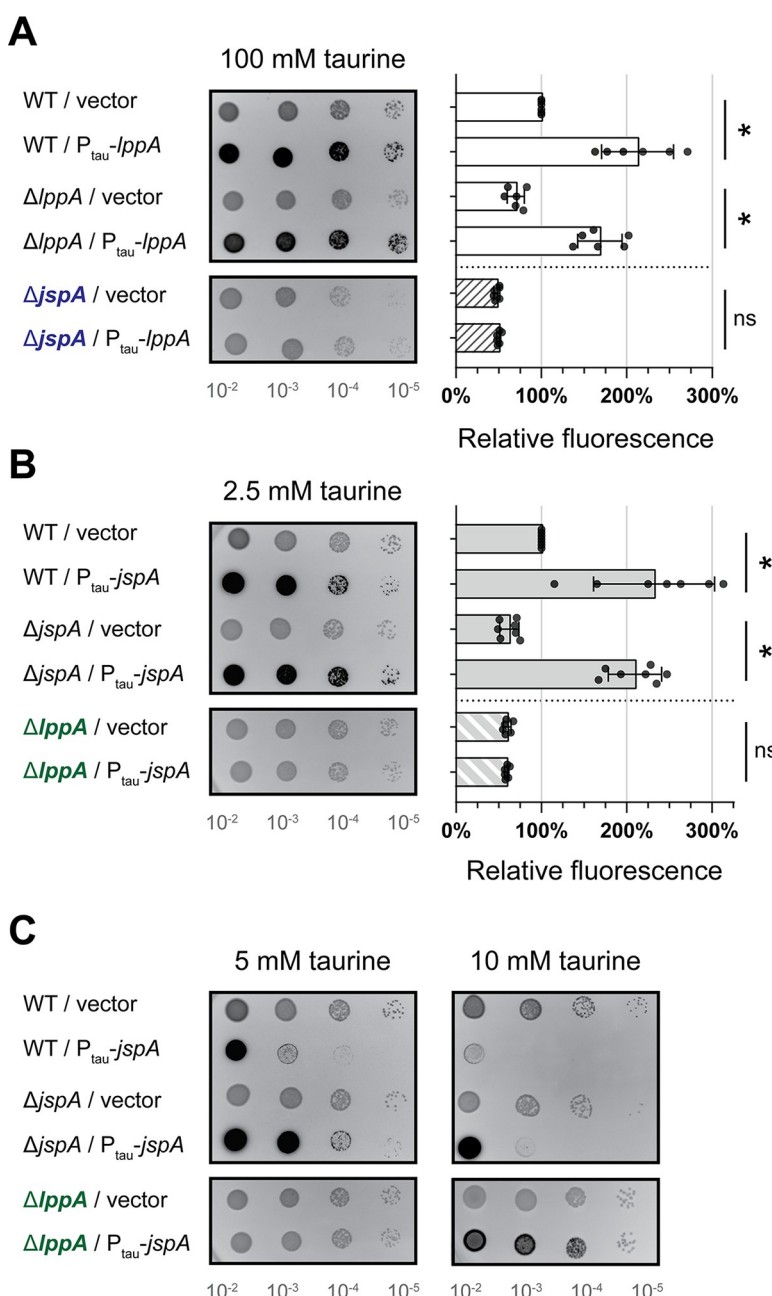

**Fig 3. Calcofluor fluorescence, indicating EPS-I production, of strains expressing *lppA* or *jspA*.** Ten-fold serial dilutions ($10^{-2}$ to $10^{-5}$) of logarithmic-phase cultures were spotted onto LB plates containing calcofluor and allowed to grow for three days prior to fluorescence imaging. Darker spots on representative images indicate brighter fluorescence. Fluorescence levels were measured relative to the *S. meliloti* Rm1021 wild-type (WT) strain carrying an empty vector on each plate. (**A**) WT strains or Δ*lppA* or Δ*jspA* mutants carrying the vector (pCM130 or pJC478) or a plasmid with *lppA* under the control of a taurine-inducible promoter (pJC532) were grown on plates containing 100 mM taurine. (**B, C**) WT strains or Δ*jspA* or Δ*lppA* mutants carrying the vector or a plasmid with *jspA* under the control of a taurine-inducible promoter (pJC535) were grown on plates containing (B) 2.5, (C) 5, or 10 mM taurine. Error bars represent standard deviations. Relative fluorescence intensities were not calculated for 5 and 10 mM taurine due to growth inhibition of select strains. *, p < 0.05; ns, not significant.

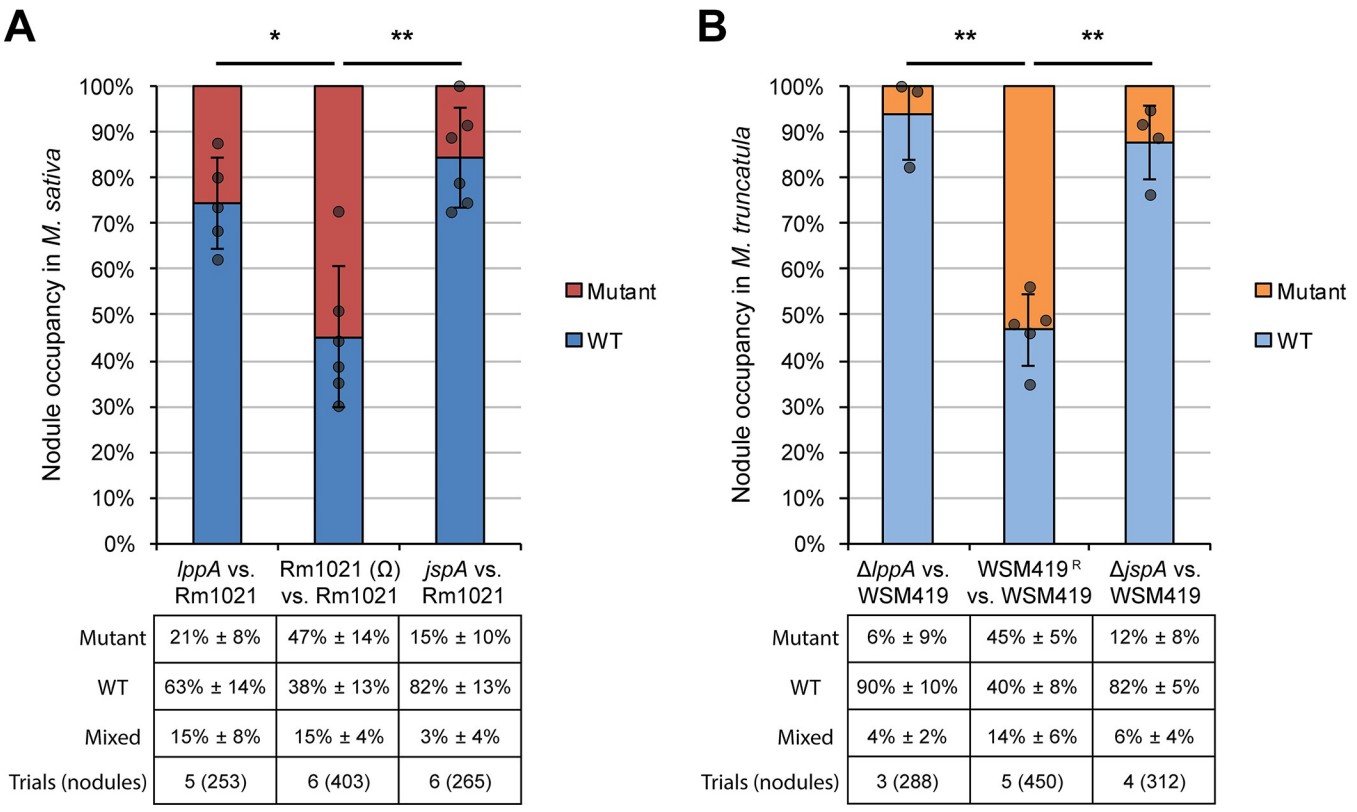

**Fig 4. Proportions of root nodules colonized by each bacterial strain after competitive infection.** The predominant strain colonizing each nodule was determined after seedlings were inoculated with equal mixtures of two strains. (**A**) *S. meliloti* Rm1021 and its derivatives were used to infect *M. sativa*, while (**B**) *S. medicae* WSM419 and its derivatives were used to infect *M. truncatula*. Rm1021 (Ω) is a derivative of Rm1021 marked with resistance to spectinomycin, while WSM419[R] are derivatives of WSM419 marked with resistance to spectinomycin or neomycin. Mutations in *jspA* or *lppA* in Rm1021 were deletions or transposon insertions, while those in WSM419 were all deletions. Percentages (± standard deviations) below each competition indicate the mean proportions of nodules containing the *jspA* or *lppA* mutant, wild type (WT), or a mixture of the two, while the graphs depict relative abundance when mixed nodules are excluded. Dark grey circles indicate the percentage of nodules occupied by WT for individual competition trials. Error bars represent standard deviations. *, $p < 0.05$; **, $p < 0.01$. Bottom row of each table [Trials (nodules)] indicates the number of trials (and total number of nodules assessed) per competition. Detailed results from the competitive symbiosis assays are available in S4 Table.

strains (marked or unmarked). In contrast, *lppA* or *jspA* mutants were recovered from significantly fewer nodules (6–26% of the nodules) when competed against wild-type strains (Fig 4 and S4 Table). These results align with previous demonstration that JspA is important for protection against the NCR247 antimicrobial peptide and for competitiveness during symbiosis between *S. meliloti* and *M. sativa* [63]. Furthermore, our results show that both JspA and LppA contribute to competitiveness, in two different model symbiotic interactions.

## JspA and LppA affect expression of EPS-I and flagellar genes

Next, we investigated how JspA and LppA may influence EPS-I production by determining if they affect gene expression in *S. meliloti*. First, we used a transcriptional fusion to the β-glucuronidase (GUS) reporter gene [36] to measure expression of *exoY*, encoding a galactosyltransferase required for EPS-I biosynthesis [49]. Expression levels were examined in both PYE and LB rich media because our past experiences indicated that differences between genotypes could be more apparent in one particular medium [44]. Consistent with EPS-I levels monitored via calcofluor fluorescence (Fig 3 and S2 Table), deletion of *jspA* or *lppA* reduced *exoY* expression to 55–72% of wild-type levels, in both LB and PYE media (Fig 5 and S5A Table).

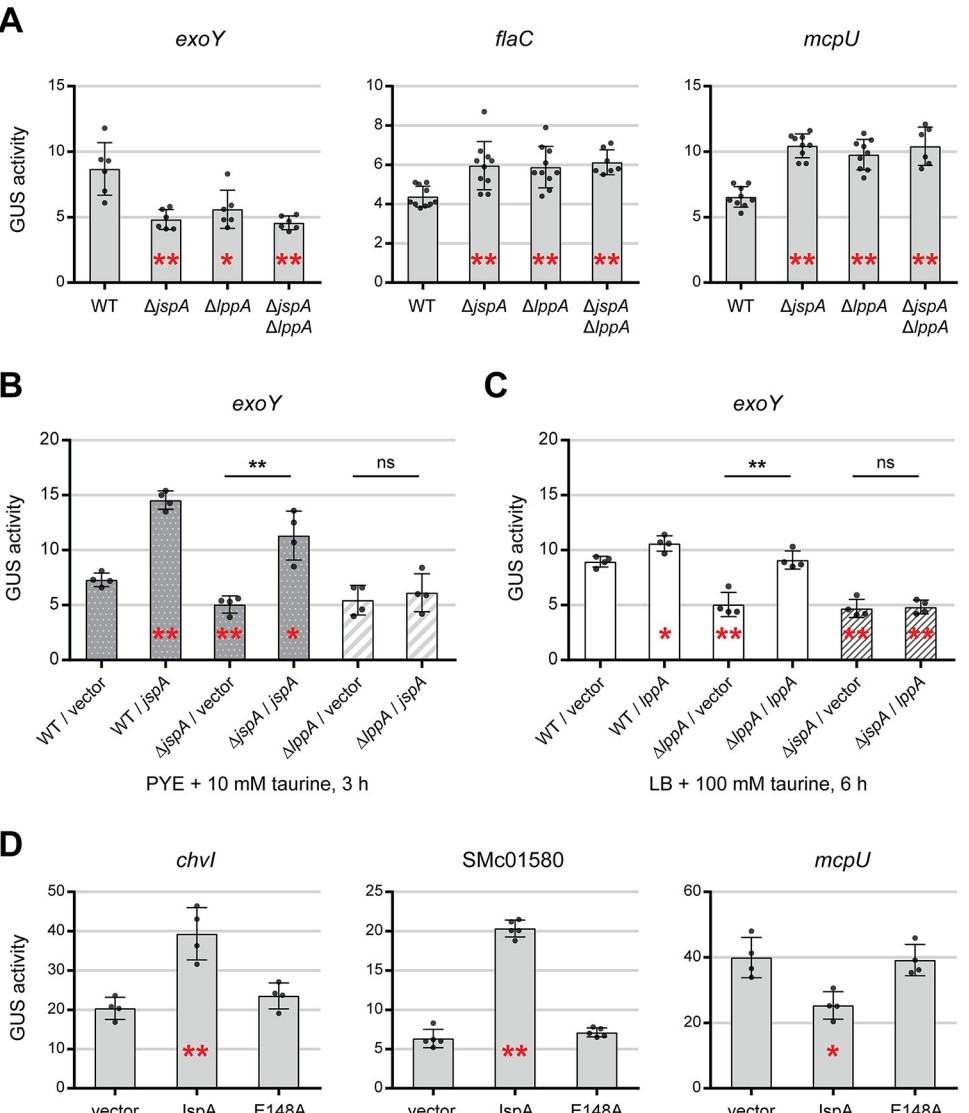

**Fig 5. Expression levels of select promoters in different genetic backgrounds.** Expression was measured using transcriptional fusions to *uidA* (encoding the GUS reporter). (**A**) Expression levels from *exoY*, *flaC*, and *mcpU* promoters in Δ*jspA*, Δ*lppA*, and Δ*jspA* Δ*lppA* mutants were compared against those in wild type (WT) grown in LB. (**B, C**) Expression from the *exoY* promoter was measured in different strains overexpressing (B) *jspA* or (C) *lppA* when grown under conditions listed below the graphs. (**D**) Expression levels from *chvI*, SMc01580, and *mcpU* promoters were assessed when JspA or JspA$_{E148A}$ was induced in PYE with 10 mM taurine for 4.5 hours. Error bars represent standard deviations. Red asterisks within bars indicate statistically significant differences when compared against the wild type or vector-bearing strain (leftmost strain in each plot), while black asterisks above bars represent significant differences between two strains under comparison: *, $p < 0.05$; **, $p < 0.01$; ns, not significant. Data for GUS reporter expression are available in S5 Table.

Second, we examined reporter fusions to *flaC* and *mcpU* [45], respectively encoding a flagellin and a chemoreceptor [89,90], as expression of genes involved in flagellar motility and chemotaxis often change in opposition to those involved in EPS-I production [44,91,92]. Deletion of *jspA* or *lppA* increased expression of these two genes significantly in LB medium (to 134–159% of wild type) (Fig 5A), but less clearly in PYE medium (to 114–129% of wild type) (S5A Table). Finally, expression levels of *exoY*, *flaC*, and *mcpU* in the Δ*jspA* Δ*lppA* double mutant were

similar to those in the single mutants (Fig 5A and S5A Table), again suggesting that JspA and LppA function in the same genetic pathway.

We also used the transcriptional fusions to *exoY* and *flaC* to assess the effects of *jspA* or *lppA* overexpression from a plasmid-borne, taurine-inducible promoter (with the same $P_{tau}$-regulated constructs as those in Fig 3). Induction of *jspA* expression with 10 mM taurine in PYE for three hours was sufficient to significantly alter expression of both *exoY* and *flaC* in the wild-type background (Fig 5B and S5B Table). Such induction also complemented the drop in *exoY* expression seen in the Δ*jspA* mutant. In contrast, induction of *lppA* expression did not affect *flaC* or *exoY* significantly under various conditions tested (S5B Table). We only found a modest increase in *exoY* transcription in the wild-type background when *lppA* expression was induced for six hours with 100 mM taurine in LB medium (Fig 5C and S5B Table). Such induction also sufficed to complement the drop in *exoY* expression in the Δ*lppA* mutant. Notably, overexpression of *jspA* could not reverse the decrease in *exoY* expression in the Δ*lppA* mutant (Fig 5B), and overexpression of *lppA* could not reverse the same effect in the Δ*jspA* mutant (Fig 5C). Overall, expression analysis with transcriptional reporters reflected the results obtained with calcofluor fluorescence (Fig 3): *lppA* appears to require higher levels of induction compared to *jspA* to cause detectable physiological changes, and both genes need each other to function efficiently.

To determine if membrane anchoring of LppA and proteolytic activity of JspA are critical for function, we mutated the lipobox motif of LppA and the peptidase domain of JspA (Fig 1B) and assessed the mutant derivatives' effects on EPS-I synthesis and gene expression. For LppA, we mutated $Cys_{23}$ of the lipobox motif to Ser and tagged both the wild-type and mutant versions at the C-terminus with an HA epitope. While constructing the *lppA-HA* allele, we serendipitously obtained alleles with conversion of $Gly_{96}$ to Trp and $Ala_{78}$ to Ser and decided to analyze each of the corresponding two mutants as well. Due to the relatively minor changes in reporter gene expression when *lppA* was overexpressed in the wild-type background (Fig 5C and S5B Table), we mainly examined the functionality of various *lppA* alleles in the Δ*lppA* background. Overexpression of these various derivatives ($LppA_{C23S}$, LppA-HA, $LppA_{C23S}$-HA, $LppA_{G96W}$-HA, $LppA_{A78S}$-HA) in the Δ*lppA* mutant from the plasmid-borne $P_{tau}$ promoter did not increase the fluorescence levels of colonies on calcofluor plates compared to the vector-only control, whereas overexpression of LppA did (Figs 6A and S2A). Induction of LppA or LppA-HA significantly elevated expression of the *exoY* fusion reporter to similar levels (180–195% of that in the Δ*lppA* mutant with the vector), while $LppA_{C23S}$ and $LppA_{C23S}$-HA did not (Fig 6B and S5C Table). The other two variants, $LppA_{G96W}$-HA and $LppA_{A78S}$-HA, increased *exoY* expression modestly but significantly (123–130%) (Fig 6B and S5C Table). Immunoblot analysis using antibodies against the HA epitope indicated that the steady-state levels of all HA-tagged derivatives of LppA, when constitutively expressed from plasmids, are relatively similar in wild-type, Δ*lppA*, and Δ*jspA* backgrounds, except for $LppA_{C23S}$-HA, which appears to be expressed at very low levels in all backgrounds (Fig 6E). These results are consistent with acylation at the Cys residue of the predicted lipobox allowing LppA to anchor to the membrane and mature into a stable lipoprotein. Addition of an HA epitope at the C-terminus appears to reduce the functionality of LppA, while the G96W and A78S mutations further diminish protein activity without affecting its stability detectably. According to the protein structure predicted by AlphaFold [93], $Gly_{96}$ is located in a hairpin loop between two beta strands, while $Ala_{78}$ is located within an alpha helix (S2B Fig). Mutations of these residues likely alter the tertiary structure and may be helpful in the future for defining the specific molecular activity of LppA.

For JspA, we mutated residues within the HEMAH active site, His147 to Ala, or Glu148 to Ala or Asp [94], and tagged wild-type and mutant versions at the C-terminus with the HA

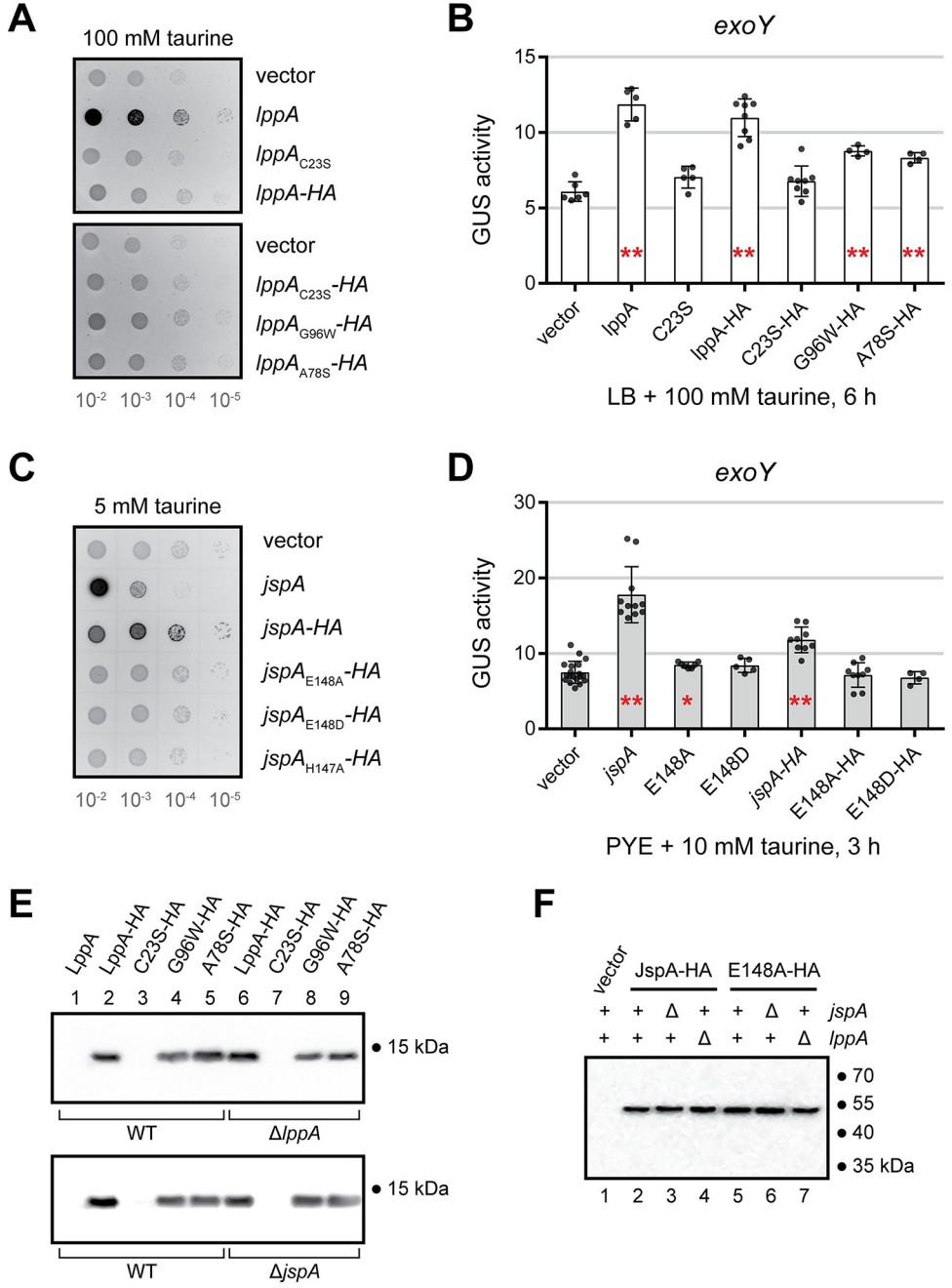

**Fig 6. Expression of mutant *lppA* and *jspA* alleles in *S. meliloti*.** (**A**) Calcofluor fluorescence was used to assess EPS-I production in Δ*lppA* mutants expressing different alleles of *lppA* from a taurine-inducible promoter. (**B**) Overexpressing different alleles of *lppA* in the Δ*lppA* mutant affected expression from the *exoY* promoter to varying degrees, as measured with transcriptional fusion to GUS. (**C, D**) Wild-type Rm1021 expressing different *jspA* alleles from a taurine-inducible promoter exhibit varying levels of (C) fluorescence on calcofluor plates and (D) expression from the *exoY* promoter. (**E**) Immunoblots show steady-state levels of different versions of HA epitope-tagged LppA in wild-type (WT), Δ*lppA*, and Δ*jspA* backgrounds. Samples were harvested from cultures grown in LB with 100 mM taurine for 6 hours. (**F**) Immunoblot shows steady-state levels of of JspA-HA and JspA$_{E148A}$-HA in different genetic backgrounds. Samples were harvested from wild type or Δ*lppA* or Δ*jspA* mutants, carrying different plasmids, grown in PYE with 10 mM taurine for 4.5 hours. Presence or absence of chromosomal *jspA* and *lppA* (+ or Δ) are indicated above each lane. Numbers to the right of immunoblots (E, F) indicate approximate molecular mass standards, in kDa. Plasmids pJC532, pJC605, pJC606, pJC607, pJC608, and pJC609 were used for expressing *lppA*, *lppA*$_{C23S}$, *lppA-HA*, *lppA*$_{C23S}$-*HA*, *lppA*$_{G96W}$-*HA*, and *lppA*$_{A78S}$-*HA*, while pJC535, pJC555, pJC556, pJC558, pJC559, pJC560, and pJC561 were used for *jspA*, *jspA*$_{E148A}$, *jspA*$_{E148D}$, *jspA-HA*, *jspA*$_{E148A}$-*HA*, *jspA*$_{E148D}$-*HA*, and *jspA*$_{H147A}$-*HA*, respectively.

Vectors used were pCM130 (A, B, D, F) or pJC478 (C). LppA and JspA variants are named according to their altered residues: for example, C23S-HA indicates LppA$_{C23S}$-HA. Error bars represent standard deviations. Red asterisks within bars indicate statistical significance when compared against the vector-carrying strain (leftmost strain in each plot): *, $p < 0.05$; **, $p < 0.01$. Data for GUS reporter expression are available in S5 Table.

epitope. When overexpressed from the plasmid-borne P$_{tau}$ promoter in the wild-type background with 5 mM taurine, both JspA and JspA-HA elevated the fluorescence of colonies on calcofluor plates, but only JspA inhibited colony formation (Fig 6C). None of the three active site mutants (H147A, E148A, E148D), untagged or tagged, enhanced fluorescence or affected growth (Figs 6C and S2C). In addition, we expressed *S. meliloti* JspA and its derivatives in three related alpha-proteobacteria—*S. medicae* WSM419, *S. fredii* NGR234, and *C. crescentus* NA1000—to assess if JspA activity is conserved (S1B and S3 Figs). In WSM419, expression of JspA complemented the EPS-I production defect of the Δ*jspA* mutant, whereas expression of JspA$_{E148A}$ did not (S1B Fig). In both WSM419 and NGR234, overexpression of JspA inhibited colony formation, while overexpression of JspA$_{E148A}$-HA did not (S3 Fig). Overexpression of JspA$_{E148A}$ and JspA-HA inhibited growth to different extents in these two species (S3 Fig), suggesting that JspA variants have different activities in distinct genetic backgrounds: overproduction of a proteolytically inactive JspA seems more deleterious in NGR234 than in *S. meliloti* Rm1021 or *S. medicae* WSM419. In contrast, expression of JspA and its derivatives in the more distantly related *C. crescentus* NA1000 did not inhibit growth until higher levels of induction (10 mM taurine), possibly due to the general stress of protein overproduction because similar levels of inhibition were observed for all alleles. These results suggest that the proteolytic function of JspA is conserved in *Rhizobiales* but not in other alpha-proteobacteria.

Results similar to those seen with calcofluor plates were obtained when analyzing JspA derivatives with transcriptional fusion of the GUS reporter to *exoY* in *S. meliloti*: overexpression of JspA-HA significantly increased *exoY* transcription, but to a lesser extent compared to untagged JspA (Fig 6D and S5C Table). The E148A and E148D variants, untagged or tagged, both failed to elevate *exoY* expression to levels achieved by JspA and JspA-HA, indicating that the active site of the protease is necessary for function (Fig 6D and S5C Table). As with JspA, JspA-HA also increased *exoY* expression in the Δ*jspA* mutant but not in the Δ*lppA* mutant, again indicating that JspA needs LppA for activity (S5C Table). Immunoblotting indicated that HA-tagged versions of JspA appear to reach similar steady-state levels in wild-type, Δ*lppA*, and Δ*jspA* backgrounds (Figs 6F and S2D), demonstrating that variations in activity are not likely due to differences in protein stability in different genetic backgrounds. Together, these results suggest that the active site of JspA is critical for its proteolytic activity, but other domains contribute to function because expression of proteolytically inactive or tagged versions of the protein can interfere with cellular processes to varying extents when overexpressed in distinct species (S3 Fig); this interpretation is consistent with previous demonstration by Arnold *et al.* [63] that the peptidase active site, the lipobox motif, and the LysM domain of JspA are all necessary for protection against the antimicrobial activity of NCR247.

## LppA and JspA participate in the ExoR-ExoS-ChvI signaling pathway

Considering that LppA and JspA are likely extracytoplasmic, we wondered how they influence transcription of EPS-I and flagellar genes. To uncover their mechanism of action, we performed whole-genome expression analysis using Affymetrix GeneChips [92]. Having generated mutant alleles and determined growth conditions and strain backgrounds with which significant changes in gene expression could be observed, we decided to examine the transcriptomes of Rm1021 carrying an empty vector or overexpressing wild-type JspA or loss-of-

function JspA$_{E148A}$ from a plasmid. As expected, overexpression of JspA caused changes in a substantial set of genes, whereas JspA$_{E148A}$ did not: pairwise comparisons for changes greater than 1.5-fold revealed 198 genes with significantly different expression between strains overexpressing JspA and carrying the vector, 155 genes between JspA and JspA$_{E148A}$, and only 5 genes between JspA$_{E148A}$ and the vector (Fig 7A and S6 Table). 141 gene expression changes were shared between the JspA versus vector and JspA versus JspA$_{E148A}$ comparisons, and consequently these genes were deemed strong candidates for the JspA transcriptome: 80 increased expression and 61 decreased expression during JspA overexpression. Consistent with measurements of *exoY* and *flaC* reporter fusions (Fig 5 and S5B Table), a sizable portion of up-regulated genes are associated with exopolysaccharide biosynthesis, while the majority of down-regulated genes are associated with flagellar motility and chemotaxis. One up-regulated target that stood out is *chvI*, encoding a conserved response regulator critical for viability and symbiosis [13,30]. To verify the results of the transcriptome analysis, we constructed transcriptional fusions of the GUS reporter to *chvI* and select candidates of the JspA transcriptome. Measurements of GUS activity showed expected increase (for *chvI*, SMc01580, and *pckA*) or decrease (for *mcpU*) when *jspA* expression is induced with taurine from the plasmid-borne P$_{tau}$ promoter, compared to the same wild type carrying the vector; in contrast, overexpression of the mutant *jspA*$_{E148A}$ allele did not elicit significant changes (Fig 5D and S5D Table). We also evaluated expression of the *chvI* reporter when JspA and variants were expressed from a plasmid-borne, IPTG-inducible promoter, P$_{lac}$ [95] to ensure that the observed changes could be replicated with another inducer. Overexpression of JspA and JspA-HA both increased *chvI* transcription, whereas JspA$_{E148A}$, untagged or tagged, did not cause similar effects (S5D Table). These reporter activities support the validity of the transcriptome analysis.

Knowing that the ExoR-ExoS-ChvI signaling system can control its own expression [30,96], we asked if JspA participates in that regulatory pathway. We compared the JspA transcriptome against the published transcriptomes of ExoR/ExoS and ChvI [21,30,36] (see Materials and methods for details) and saw substantial overlap among the three sets of genes (Fig 7B). In contrast, the JspA transcriptome had minimal overlap with the published transcriptome of RpoH1, a heat shock sigma factor (Fig 7C) [97], chosen for comparison because it represented a stress response distinct from that of the ExoR-ExoS-ChvI system. Hypergeometric probability tests [98] indicated that overlap among the JspA, ExoR/ExoS, and ChvI transcriptomes are highly significant, whereas each of the three sets overlapped poorly with the RpoH1 transcriptome (S7 Table). As noted previously [30], there are significant overlaps between the ChvI regulon and the groups of genes affected by the *podJ1* mutation [44] or by NCR247 treatment [40]. To be expected for one sharing the same genetic pathway as ChvI, JspA's transcriptome also intersects significantly with the *podJ1* and NCR247 sets (S8 Table). These similarities suggest that JspA contributes to a regulatory pathway, likely the ExoR-ExoS-ChvI system, for responding to specific stress conditions, such as those caused by the *podJ1* mutation or exposure to NCR247.

To determine if JspA and LppA influence the ExoR-ExoS-ChvI signaling pathway, we conducted epistasis analysis, first using Tn5 insertions in *exoR* and *exoS* that lead to overproduction of EPS-I [27]. Mutants carrying the *exoS96*::Tn5 insertion produce an N-terminally truncated ExoS that behaves like a constitutively active kinase [13], while mutants carrying the *exoR95*::Tn5 insertion produce a C-terminally altered ExoR that has lost function [23]. Loss of *lppA* or *jspA* in these backgrounds did not reduce EPS-I synthesis, suggesting that *exoR* and *exoS* are epistatic to *lppA* and *jspA* (Fig 8A and S2C Table). Considering that, like ExoR, mature JspA and LppA are predicted to reside in the periplasm, and ExoR inhibits ExoS-ChvI signaling, we hypothesized that JspA and LppA together negatively regulate ExoR activity (Fig 1). This model is consistent with the loss of JspA or LppA being unable to reduce EPS-I production if ExoR is inoperative or if ExoS is constitutively active.

## A Altered gene expression when JspA overexpressed
> 1.5-fold change

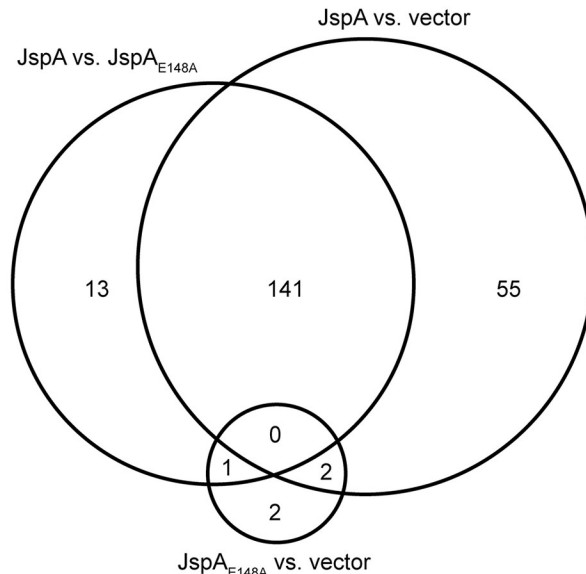

**80 genes UP-regulated**

| | |
|---|---|
| Exopolysaccharide biosynthesis: | 23 |
| Regulators (including ChvI): | 6 |
| Metabolism: | 6 |
| Miscellaneous: | 10 |
| Hypothetical proteins: | 35 |

**61 genes DOWN-regulated**

| | |
|---|---|
| Flagellar biosynthesis and motility: | 40 |
| Pili biogenesis: | 2 |
| Metabolism and biosynthesis: | 9 |
| Regulators: | 3 |
| Miscellaneous: | 1 |
| Hypothetical proteins: | 6 |

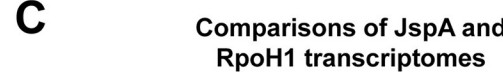

## B Comparisons of JspA and ExoR/ExoS transcriptomes and ChvI regulon

## C Comparisons of JspA and RpoH1 transcriptomes

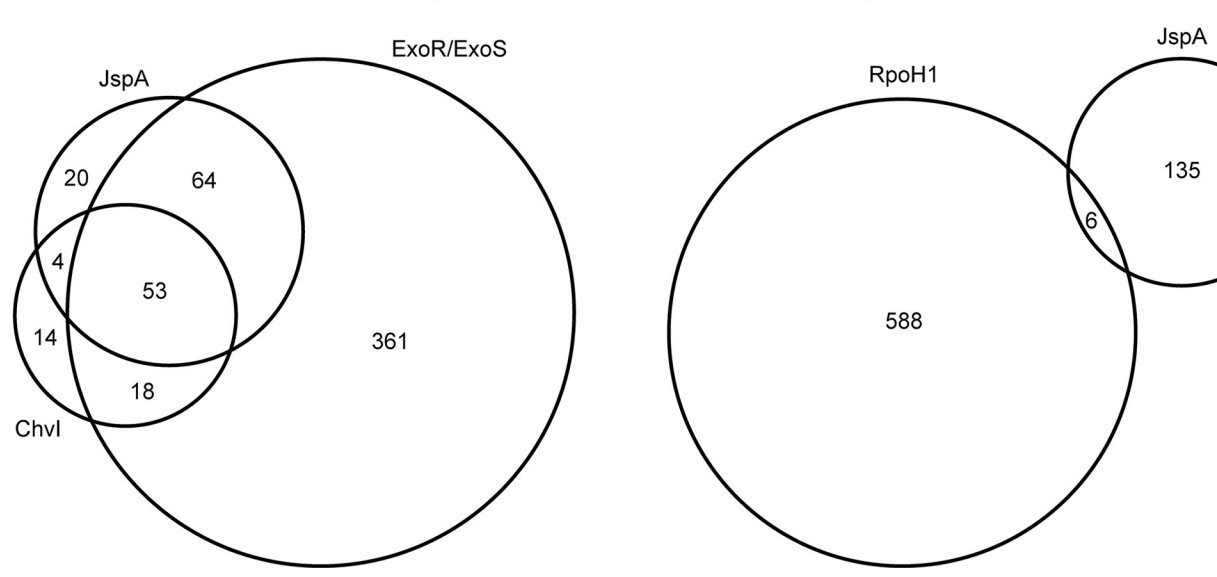

**Fig 7. Venn diagrams depicting overlaps of gene sets.** (**A**) Circles of the Venn diagram represent the numbers of genes whose expression changed >1.5-fold in three pairwise comparisons: between strains that overexpress wild-type JspA or mutant JspA$_{E148A}$ (JspA vs. JspA$_{E148A}$), between strains that overexpress JspA or carry the vector pCM130 (JspA vs. vector), or between strains that overexpress mutant JpsA or carry the vector (JspA$_{E148A}$ vs. vector). The 141 genes that appeared in both the JspA vs. vector and JspA vs. JspA$_{E148A}$ comparisons were grouped according to their functions, as listed on the right. ChvI belongs to the group of regulators whose gene expression increased when JspA was overexpressed. (**B, C**) The bottom Venn diagrams represent the overlaps of (B) genes that belong to the JspA or ExoR/ExoS transcriptome or ChvI regulon and (C) those that belong to the RpoH1 or JspA transcriptome. Gene sets and analyses of their overlaps are provided in S6 and S7 Tables. See Materials and methods for details about assignment of genes to the ChvI regulon and ExoR/ExoS transcriptome.

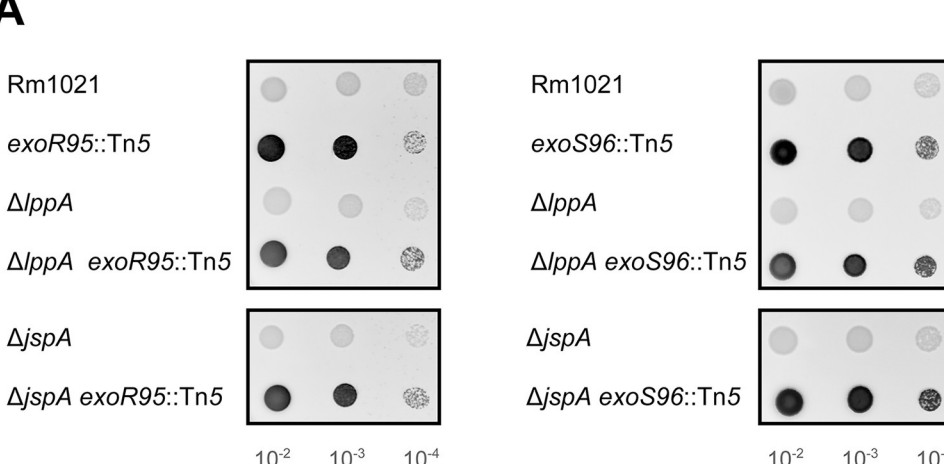

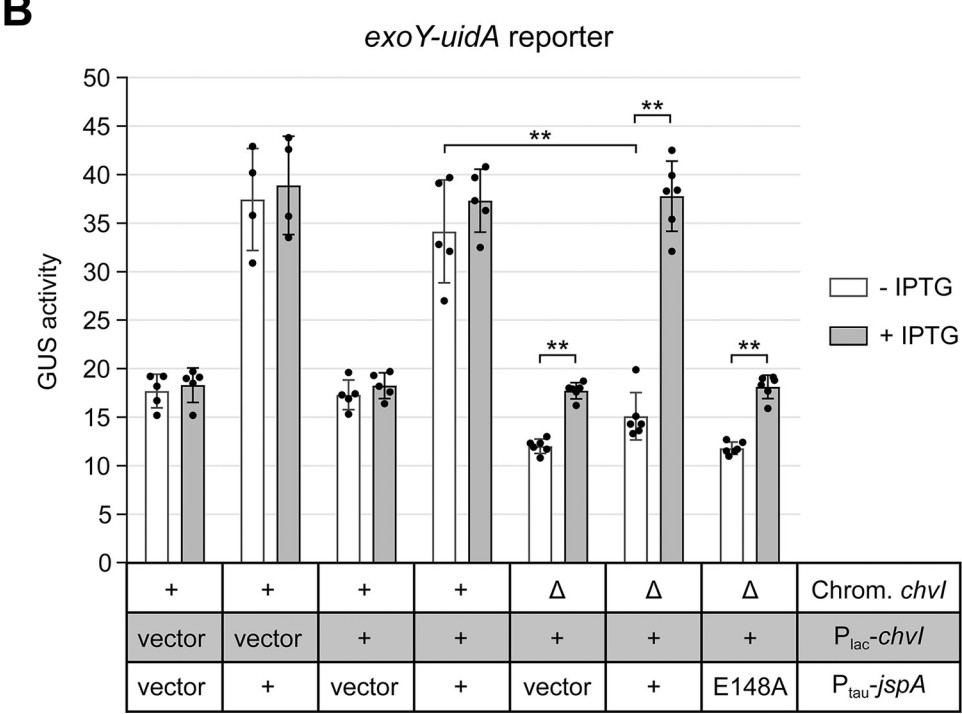

**Fig 8. Epistatic interaction between *jspA* and *lppA* and the *exoR-exoS-chvI* pathway.** (**A**) Calcofluor fluorescence of wild-type and mutant strains were assessed by spotting ten-fold serial dilutions of cultures onto LB plates. Strain genotypes are shown to the left of the fluorescence images. Representative images are shown, and at least two replicates were included for each comparison. Measurements of relative fluorescence are available in S2C Table. (**B**) Expression of the *exoY-uidA* reporter was monitored in strains replete with or depleted of ChvI, while *jspA* or *jspA*$_{E184A}$ was ectopically expressed. Relevant alleles on the chromosome and on plasmids are indicated below the plot: first row indicates the presence or absence of chromosomal *chvI* (+ or Δ), second row indicates presence of empty vector or a plasmid that expresses *chvI* (vector or +), and the third row indicates presence of vector or a plasmid that expresses wild-type or mutant *jspA* (vector, +, or E148A, respectively). Strains with the Δ*chvI* allele (rightmost three strains) carried a complementing plasmid (pAD101) with *chvI* under the control of the P$_{lac}$ promoter (P$_{lac}$-*chvI*): growth in the presence or absence of IPTG resulted in expression or depletion of ChvI. For comparison, *chvI*$^+$ strains carried the P$_{lac}$-*chvI* plasmid or the corresponding parent vector (pSRKKm). All the strains also bore a compatible vector (pCM130) or its derivatives that enable taurine-regulated expression of *jspA* or *jspA*$_{E184A}$ (pJC535 or pJC555). Strains were grown in LB with 10 mM taurine for 6 hours, while expressing or depleting ChvI, prior to measurement of GUS activities. Averages and standard deviations (error bars) were calculated with measurements from at least four different days (S5E Table). \*\*, $p < 0.01$ between specified measurements.

To test this idea further, we constructed a ChvI depletion strain, in which the only copy of *chvI* is under the control of $P_{lac}$ on a pBBR1-derived plasmid [95]. As ChvI is essential for growth on rich medium [17,18], the ChvI depletion strain grew normally in the presence of the IPTG inducer, similar to a *chvI*$^+$ strain carrying the same plasmid grown with or without IPTG, and poorly in the absence of the inducer (S4 Fig and S9A Table). We then monitored expression of the *exoY* reporter when ChvI was replete or depleted and when wild-type or mutant JspA was overexpressed from the $P_{tau}$ promoter on a compatible RK2-derived plasmid [83] (Fig 8B and S5E Table). In a *chvI*$^+$ strain carrying the $P_{lac}$ vector or the $P_{lac}$-*chvI* plasmid, constitutive expression of JspA from $P_{tau}$ increased *exoY* expression compared to the same strain carrying the $P_{tau}$ vector (Fig 8B, first four strains on the left), consistent with previous measurements (Fig 5B and S5B Table). In the ChvI depletion strain carrying the $P_{tau}$ vector, shutting off *chvI* expression by removing IPTG for six hours reduced *exoY* expression (Fig 8B, fifth strain from left). When JspA was overexpressed in the ChvI depletion strain, *exoY* expression was high when ChvI was replete in the presence of IPTG (Fig 8B, sixth strain from left, + IPTG), comparable to that seen in *chvI*$^+$ strains when JspA was overexpressed. However, depletion of ChvI in the absence of IPTG prevented *exoY* expression from becoming elevated by JspA (Fig 8B, sixth strain from left, - IPTG). A ChvI depletion strain overexpressing JspA$_{E148A}$ (Fig 8B, rightmost strain) yielded similar *exoY* expression patterns as the depletion strain carrying the $P_{tau}$ vector (Fig 8B, fifth strain from left). These results support the model that JspA functions upstream of ChvI: increasing JspA levels relieves the inhibitory activity of ExoR, in turn activating the ExoS sensor kinase and ChvI response regulator and promoting expression of EPS-I genes and thus EPS-I production. In the absence of ChvI, JspA is unable to stimulate expression of EPS-I genes, such as *exoY* (Fig 1).

The ExoR-ExoS-ChvI system effects changes in gene expression in response to environmental conditions, including acid stress [99–101]. Since JspA and LppA appear to act upstream of the system, we investigated if they mediate transmission of environmental signals. We compared the JspA transcriptome and ChvI regulon against sets of genes that changed expression upon acid stress, identified in three different studies [37–39]. Hypergeometric probability tests indicate that the overlaps between the JspA transcriptome or ChvI regulon and each of the three sets of acid response genes are significant but not as strong as that between JspA and ChvI: the most significant overlaps with the JspA transcriptome and ChvI regulon belong to the set identified by Hellweg, Pühler, and Weidner [38] (S10 Table). However, the overlaps between the JspA transcriptome or ChvI regulon with each of the three sets of acid response genes are comparable to, if not better than, overlaps among the three. Thus, we chose four representative genes (SMb21188, SMc01580, *exoY*, and *chvI*) from the JspA transcriptome and ChvI regulon that also appeared in one or more of the acid responses and examined their expression via reporter fusions in acidic, neutral, or basic pH. All four reporter fusions increased expression when wild-type Rm1021 was grown at pH 6 compared to pH 7; only *exoY* showed significant increase at pH 8.5 as well (Fig 9 and S5F Table). Deletion of *jspA* or *lppA* appeared to curtail this increase in response to acid stress, more obviously for SMb21188 and SMc01580 (Fig 9A and 9B, pink bars). For *exoY* and *chvI*, the deletions reduced reporter expression compared to wild type at neutral pH (Fig 9C and 9D, yellow bars), and the fold-change between pH 6 and pH 7 in the deletion mutants approximated that seen in wild type (Fig 9C and 9D, red percentages). Nevertheless, in the deletion mutants, the increase in *exoY* expression due to growth at pH 6 was impacted more severely than the increase due to growth at pH 8.5 (Fig 9C). Our results suggest that, while factors other than ChvI may help regulate *exoY* and *chvI* expression upon acid stress, JspA and LppA facilitate ChvI's response to acid stress.

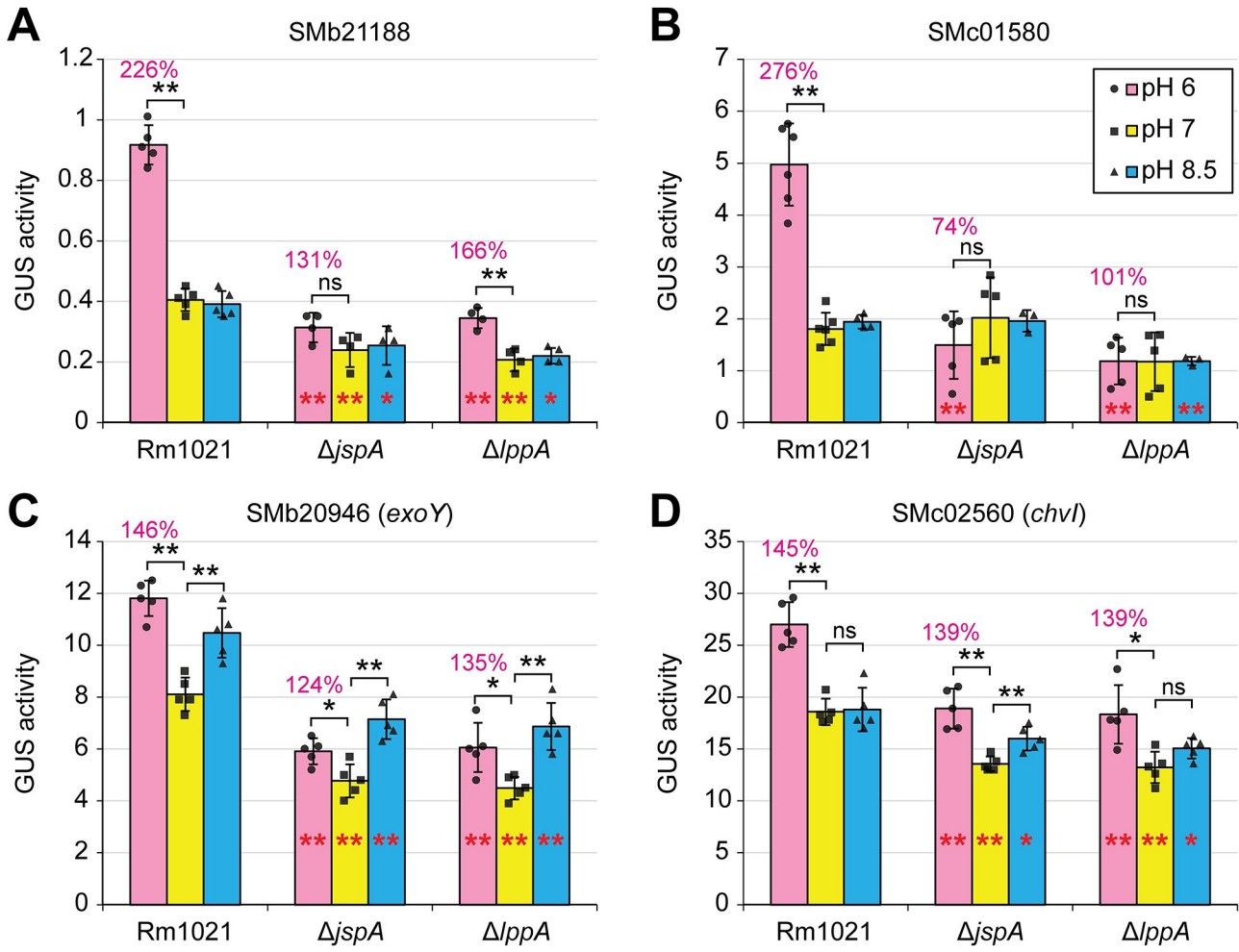

**Fig 9. Effects of *jspA* and *lppA* on transcriptional responses to pH shift.** Changes in gene expression were determined using GUS fusions to (**A**) SMb21188, (**B**) SMc01580, (**C**) SMb20946 (*exoY*), and (**D**) SMc02560 (*chvI*) at their native loci, generated in such a way as to preserve the function of the gene being examined. GUS activities in wild-type (Rm1021), Δ*jspA*, and Δ*lppA* backgrounds were measured 4.5 hours after cultures were shifted from pH 7 to pH 6 (pink bars), 7 (yellow bars), or 8.5 (blue bars) in LB medium. Activity at pH 6 relative to pH 7 for each genotype is shown as the red percentage above each pink bar. Maroon * or ** within a bar for one of the mutants represents significant difference (p < 0.05 or p < 0.01, respectively) when compared to the same condition in wild type. Analogously, black * or ** above the bars indicates significant difference when activity at pH 6 or 8.5 is compared to that at pH 7 for the same genotype, while ns indicates no significant difference. Averages and standard deviations (error bars) were calculated from three to six independent measurements (S5F Table).

## JspA and LppA enhance ExoR degradation

Because JspA is predicted to be a periplasmic protease, and JspA and LppA appear to promote ExoS/ChvI activity, in opposition to ExoR, which can be regulated via proteolysis [23,25], we assessed whether JspA reduces ExoR levels. We introduced a plasmid expressing *jspA* or *jspA*$_{E148A}$ from P$_{lac}$ into a derivative of Rm1021 with *exoR-V5* instead of *exoR* at the native locus. Induction of *jspA* expression with 1 mM IPTG in rich media inhibited growth, starting 3–4 hours after induction (Figs 10A and S5 and S9 Table). Overexpression of *jspA*$_{E148A}$ did not retard growth compared to a strain carrying the P$_{lac}$ vector when cultures were grown in flasks (Figs S5B and S5C and S9 Table) but did slow growth in 48-well plates (Figs 10A and S5A), such that the doubling time of the strain overexpressing *jspA*$_{E148A}$ was seven hours, compared to six hours for a strain carrying the vector (S9B Table). This difference in growth between

**A**

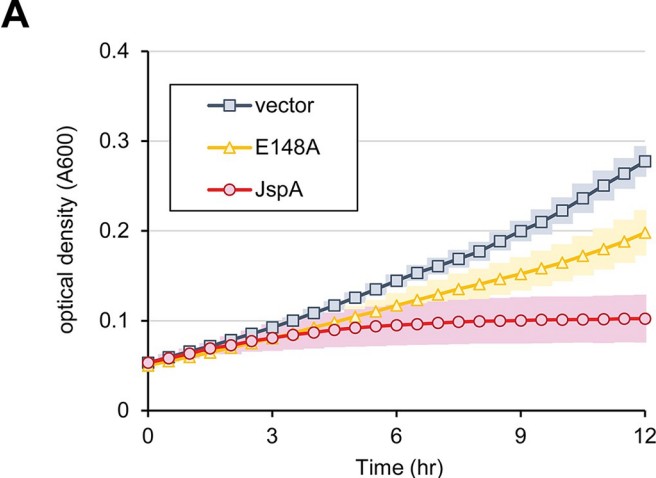

**B**

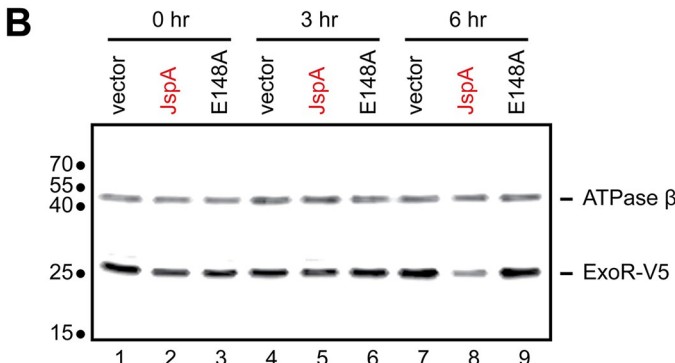

**C**

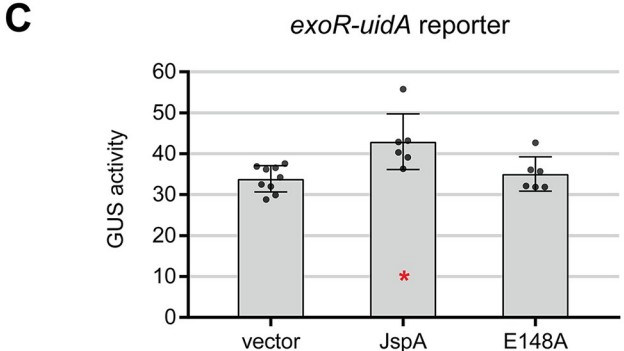

**Fig 10. Effects of JspA on ExoR levels.** (**A**) Plot depicts growth of *exoR-V5* strains carrying the pSRKGm vector or derivatives (pJC652 or pJC653) with *jspA* or *jspA*$_{E184A}$ (noted as E148A) under control of the P$_{lac}$ promoter. Strains JOE5242, JOE5244, and JOE5246 were grown in 48-well plates, with 0.4 mL PYE plus 1 mM IPTG per well. Absorbance at 600 nm (A600) was measured every 30 minutes. Average readings for three different days are depicted, with surrounding shadings indicating standard deviations. In the absence of IPTG, all strains exhibited growth patterns similar to that of the vector-carrying strain in the presence of IPTG (see S5A Fig). (**B**) Immunoblot shows steady-state levels of ExoR-V5 and the beta subunit of ATP synthase at 0, 3, and 6 hours after inducing expression of *jspA* or *jspA*$_{E184A}$, compared against levels in the vector-carrying strain. Approximate molecular mass, in kDa, are shown to the left of the blot, while lane numbers are shown below. Growth conditions were similar to that in (A), except that strains were cultured in flasks. (**C**) Expression of the *uidA* reporter fusion to *exoR* from its native locus was assessed in strains carrying the vector (pCM130) or overproducing JspA or JspA$_{E148A}$ (with pJC535 or pJC555). Cultures were grown with 10 mM taurine for 4.5 hours prior to measurement of GUS activities. Averages and standard deviations were calculated from at least four measurements (S5G Table). *, p < 0.05 when compared against the vector-carrying strain.

culture formats may be due to variations in aeration, trace contaminants in the flasks, or other unknown factors. For cultures grown in flasks, we monitored steady-state levels of ExoR-V5 by immunoblotting with antibodies against the V5 epitope, at zero, three, and six hours after induction. ExoR-V5 level was detectably lower in the strain overexpressing JspA compared to strains carrying the vector or expressing mutant $JspA_{E148A}$, six hours after induction (Fig 10B). Our transcriptomic analysis indicated that *exoR* expression is slightly elevated (1.3x) (S6 Table) when *jspA* is constitutively induced. This increase in *exoR* expression was verified using transcriptional fusion to the GUS reporter (Fig 10C and S5G Table). Thus, the decrease in ExoR level is not attributable to a drop in *exoR* expression. Instead, JspA appears to negatively regulate ExoR at the protein level. The reduction in ExoR likely stimulates the ExoS-ChvI pathway, resulting in feedback that elevates *exoR* transcription, as previously described [96].

To eliminate changes in ExoR levels due to transcriptional regulation, we placed a FLAG-tagged version of *exoR* under the control of the $P_{lac}$ promoter on a plasmid and induced expression with 0.5 mM IPTG. Immunoblotting with anti-FLAG antibodies revealed a band for the mature ExoR-FLAG protein at ~29 kDa, as well as a band with slightly larger molecular mass, indicative of the pre-processed form (computed to be 32 kDa with the signal peptide) (Fig 11A, lane 2). We also observed additional bands with smaller molecular masses, likely representing degradation products, the most prominent one (labeled as "deg" in Fig 11) being approximately 20-kDa in size and possibly the same as a C-terminal fragment of ExoR ($ExoR_{C20}$) detected in a previous study [23]. Co-expression of JspA variants altered the steady-state levels of mature ExoR-FLAG to different extents, whereas levels of the ExoR-FLAG precursor remained relatively uniform, consistent with ExoR being degraded in the periplasm. In the $jspA^+$ background, overexpression of JspA-HA reduced the steady-state level of mature ExoR-FLAG, compared to when the strain carried the vector or expressed mutant $JspA_{E148A}$-HA (Fig 11A, lanes 2–4). In the $\Delta jspA$ background, ExoR-FLAG levels were elevated compared to wild type, and overexpression of JspA-HA reduced that elevation, while $JspA_{E148A}$-HA did not (Fig 11A, lanes 6–8). Probing with antibodies against the HA epitope indicated that steady-state levels of JspA-HA and $JspA_{E148A}$-HA were comparable. Expression of untagged versions of JspA and JspAE148A in both the $\Delta jspA$ and $jspA^+$ backgrounds (Fig 11A, lanes 10–13) led to similar effects as the corresponding HA-tagged variants on ExoR-FLAG, indicating that the tagged and untagged versions of JspA behaved similarly in this assay. Next, we examined steady-state levels of ExoR-FLAG when LppA is present or absent. Again, overexpression of JspA-HA reduced ExoR-FLAG levels in both $jspA^+$ and $\Delta jspA$ backgrounds (Fig 11B, lanes 1 and 3), compared to the same strains expressing mutant $JspA_{E148A}$-HA (Fig 11B, lanes 2 and 4). In contrast, in the $\Delta lppA$ background, overexpression of JspA-HA did not reduce ExoR-FLAG levels compared to overexpression of $JspA_{E148A}$-HA (Fig 11B, lanes 5 and 6). These results reinforce that JspA and LppA concertedly regulate the ExoR-ExoS-ChvI pathway by reducing ExoR protein levels.

To demonstrate further that JspA and LppA participate in ExoR proteolysis, we expressed the three proteins in two heterologous systems, *C. crescentus* NA1000 and *E. coli* DH10B, both of which lack clear *lppA* and *exoR* orthologs and contain weak *jspA* orthologs (S1 Table). In *C. crescentus* NA1000, ExoR-FLAG was expressed from an IPTG-inducible $P_{lac}$ promoter on a pBBR1-based plasmid, while different combinations of JspA variants and LppA were co-transcriptionally expressed from a $P_{tau}$ promoter on a compatible RK2-derived plasmid. When ExoR-FLAG was expressed in the absence of JspA or LppA, we detected both the mature and pre-processed forms of the protein, as well as various degradation products, by immunoblotting with anti-FLAG antibodies (Fig 11C, lane 2). Expression of JspA, but not mutant $JspA_{E148A}$, reduced the steady-state level of mature ExoR-FLAG but did not affect the pre-processed form (Fig 11C, lanes 3 and 4). Levels of mature ExoR-FLAG dropped further when

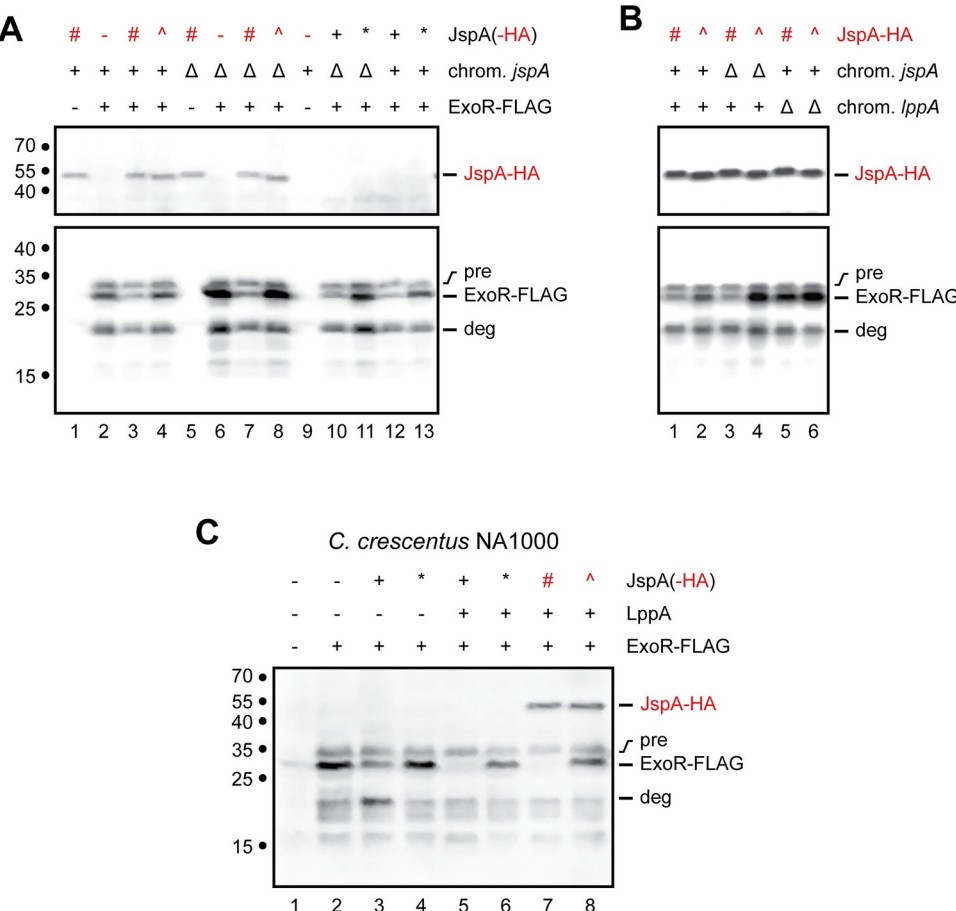

**Fig 11. Steady-state levels of ExoR-FLAG when *jspA* or *lppA* differentially expressed.** (**A**) Levels of ExoR-FLAG in the presence of different versions of JspA were assessed by immunoblotting with anti-FLAG antibodies (bottom blot), while expression of JspA-HA was verified with anti-HA antibodies (top blot). ExoR-FLAG expression is indicated above the blots: + signifies that expression of ExoR-FLAG from pMB859 was induced with 0.5 mM IPTG in TY medium for 4.5 hours, while—signifies that the strain carried the empty vector pSRKKm under the same conditions. Presence (+) or deletion (Δ) of the native *jspA* in the chromosome is also indicated. Different versions of JspA were induced with 10 mM taurine as follows: wild-type JspA from pJC614 (black +), mutant JspA$_{E148A}$ from pJC615 (black *), wild-type JspA-HA from pJC616 (red #), mutant JspA$_{E148A}$-HA from pJC617 (red ^), and no expression from the vector pJC473 (-). (**B**) Immunoblots show steady-state levels of ExoR-FLAG and JspA-HA in the presence or absence of chromosomal *lppA*. The presence (+) or deletion (Δ) of native *jspA* or *lppA* on the chromosome is shown above the blots. Expression of JspA-HA, wild-type (#) or the JspA$_{E148A}$ mutant (^), is indicated above the blots as in (A) (JspA-HA row). All strains in (B) expressed ExoR-FLAG from pMB859, induced with 0.5 mM IPTG. (**C**) Levels of ExoR-FLAG were assessed in *C. crescentus* NA1000 when JspA variants and LppA were co-expressed. Expression of ExoR-FLAG, LppA, and JspA is indicated above the blot for each lane, with—denoting no expression, and + denoting expression; *, #, and ^ denote expression of JspA$_{E148A}$, JspA-HA, and JspA$_{E148A}$-HA, respectively. ExoR-FLAG was induced from pMB859 with 0.1 mM IPTG in PYE medium for 4 hours, while a strain carrying the empty vector pSRKKm was used for no ExoR-FLAG expression. Expression of LppA and different versions of JspA were induced with 10 mM taurine using the plasmids: lanes 1 and 2, pJC473 vector; lane 3, pJC614 (JspA); lane 4, pJC615 (JspA$_{E148A}$); lane 5, pJC702 (LppA and JspA); lane 6, pJC706 (LppA and JspA$_{E148A}$); lane 7, pJC707 (LppA and JspA-HA); and lane 8, pJC708 (LppA and JspA$_{E148A}$-HA). Approximate molecular mass, in kDa, are shown to the left of the blots, while lane numbers are shown below. Positions of bands representing JspA-HA and ExoR-FLAG are indicated to the right of the blots: "pre" indicates the precursor form of ExoR-FLAG, "ExoR-FLAG" the mature form, and "deg" a major degradation product. All blots were first probed with anti-FLAG antibodies and then anti-HA antibodies (see Materials and methods). For blots in (A) and (B), the anti-FLAG images were captured first, while the anti-HA images were acquired from the same respective blots after the second probing. For the blot in (C), the image was captured after the second probing.

LppA was expressed along with JspA or JspA-HA (Fig 11C, lanes 5 and 7); this reduction did not happen when LppA was expressed with untagged or tagged versions of JspA$_{E148A}$ (Fig 11C, lanes 6 and 8). Intriguingly, one of the ExoR-FLAG degradation products, approximately 20-kDa in size and likely the ExoR$_{C20}$ fragment previously noted [23], became more prominent when only JspA, untagged or HA-tagged, was expressed (Fig 11C, lane 3; S6 Fig, lanes 5 and 9), but lost this prominence when LppA was co-expressed with JspA (Fig 11C, lanes 5 and 7; S6 Fig, lanes 7 and 11), suggesting that LppA facilitates complete degradation of ExoR.

Similar results were observed in *E. coli* DH10B, in which ExoR-FLAG was again expressed from the same P$_{lac}$ promoter on a pBBR1-based plasmid, while expression of wild-type or mutant JspA-HA and varying levels of LppA-HA was achieved by placing different constructs under the control of a weakened P$_{trc}$ promoter on a pBR322-based plasmid [102] (see Materials and methods) (S7 Fig and S1 File). In DH10B, expression of JspA-HA alone did not reduce the steady-state level of mature ExoR-FLAG (S7 Fig, lane 2), but co-expression of JspA-HA and LppA-HA did (S7 Fig, lanes 4 and 6). Furthermore, the extent of reduction in ExoR-FLAG levels depended on the level of LppA-HA expression, such that mature ExoR-FLAG became undetectable when LppA-HA was highly overexpressed (S7 Fig, lane 8). This decrease in ExoR-FLAG levels failed to occur in the presence of mutant JspA$_{E148A}$-HA (for example, lane 7 in S7 Fig). These degradation patterns in *C. crescentus* and *E. coli* suggest that LppA assists JspA to proteolyze ExoR.

## Discussion

In this report, we demonstrated that two lipoproteins, JspA and LppA, jointly contribute to the production of EPS-I by regulating expression of relevant biosynthesis genes. Each also contributes to competitiveness in nodule colonization during symbiosis with *Medicago* hosts. Site-directed mutagenesis indicated that the lipobox motif of LppA and active site residues of the JspA protease are critical for their functions, consistent with annotations of predicted domains. Transcriptome, epistasis, and Western blot analyses further revealed that the two lipoproteins influence signaling through the conserved ExoS-ChvI two-component pathway and modulate the steady-state levels of ExoR, a periplasmic inhibitor of ExoS. Exposure to acidic pH is a potential cue for activating the signaling pathway.

These results suggest a model in which JspA and LppA respond to cell envelope stress, such as exposure to acidic pH, and facilitate the degradation of ExoR, thus enhancing phosphorelay in the ExoS-ChvI system, which generates physiological changes to counter the stress (Fig 1). This regulation via proteolysis appears analogous to how *E. coli* and other Gram-negative bacteria respond to envelope stress with the Cpx and sigma(E) pathways [103,104]. For the Cpx response, accumulation of misfolded proteins in the periplasm can cause DegP to degrade CpxP, a periplasmic inhibitor of the CpxAR two-component system [105]. For the sigma(E) response, unfolded outer membrane proteins activate a proteolytic cascade involving DegS and RseP to degrade RseA, an inner membrane anti-sigma factor that inhibits sigma(E) [106,107]. While the exact molecular signal that induces degradation of ExoR is unknown, our results (Fig 9) and evidence from other alpha-proteobacteria [99,101] indicate that acidic pH can activate the ExoS-ChvI pathway, possibly by causing protein misfolding in the cell envelope. This succinct interpretation is complicated by the observations that genes that change expression in response to acid stress do not correspond precisely with those in the JspA transcriptome or ChvI regulon in *S. meliloti* (S10 Table), and that some genes can respond to acid stress in the absence of JspA or LppA (Fig 9). Most likely, other cell envelope perturbations can potentiate the stress signal, and partially redundant pathways can induce relevant physiological responses as well. Nevertheless, this model can accommodate a number of scenarios for

how JspA and LppA jointly respond to envelope stress: for instance, LppA may enable the proper folding or positioning of JspA in the membrane, or misfolding of LppA may directly induce JspA's proteolytic activity. In particular, the complex physical properties of the outer membrane [108,109] necessitate multiple regulatory checkpoints, and LppA may sense its integrity, analogous to the *E. coli* RcsF lipoprotein [110], and transmit disturbances to JspA. Alternatively, LppA may monitor or participate in the crosslink between the outer membrane and the cell wall [111], and disruptions in the process are relayed to JspA. Other envelope proteins that influence ExoS-ChvI signaling, such as SyrA [43], may also participate in the activation. Further investigation to elucidate the precise mechanisms involved would advance understanding of stress response in *Rhizobiales*.

Notably, the *jspA* and *lppA* genes were originally identified in a suppressor analysis of a *podJ1* mutant, which exhibits pleiotropic defects in the cell envelope [44]: while the *podJ1* mutant grew poorly on LB medium with low salt concentrations, null mutations in *jspA* or *lppA* alleviated the growth defect. A subdued envelope stress response when JspA or LppA is absent may allow better growth of the *podJ1* mutant under specific conditions, as too much activation can be deleterious. This interpretation is consistent with the suggestion that stress response requires careful management to avoid toxicity [103]. For example, deletion of *rseA* in *E. coli* causes constitutive activation of the sigma(E) system, resulting in membrane defects associated with lethality in stationary phase [112]. Similarly, loss of *exoR* in *S. meliloti* led to lethality, or at least severely thwarted growth [63,113–115], just as overexpression of *jspA* did in the present study, presumably due to hyperactivation of the ExoS-ChvI pathway. Deletion of *exoR* does not appear to retard growth as strongly in related species such as *B. abortus* and *A. tumefaciens* [101,116], consistent with ExoS and ChvI being critical for growth in *S. meliloti* [17,18] but not in these two other genetic models [15,20]. Whether *exoR*, *exoS*, and *chvI* orthologs are required for viability appears to vary in other members of the *Rhizobiales* as well [117–119]. This variability in the impact of conserved signaling pathways is not unprecedented. For example, sigma(E) is essential in *E. coli* [120] but not in *S. typhimurium* [121,122].

Effective management of envelope perturbations allows adaptation to environmental changes, including those encountered during symbiosis (from mutualistic to pathogenic) [6,123]. As impairment of ExoS and ChvI disrupts symbiosis [17,18,21,26,43], and their orthologs are required for virulence in *A. tumefaciens* and *B. abortus* [15,20], *S. meliloti* likely modulates ExoS-ChvI signaling to promote gene expression patterns conducive to invasion and persistence within a eukaryotic host [21,30], akin to pathogenic Gram-negative bacteria that use the Cpx and sigma(E) pathways to express virulence factors to ensure survival during infection [104,123–125]. For instance, JspA and LppA may contribute to competitiveness by ensuring an appropriate degree of EPS-I production, as the level of symbiotic EPS can optimize interaction with plant hosts [52].

Other genes regulated by JspA likely also contribute to efficient symbiosis. For example, JspA inhibits expression of the transcription regulator LdtR, which plays a role in osmotic stress tolerance, motility, and likely cell wall remodeling [126,127]. JspA increases expression of *lsrB*, which encodes a LysR-family transcription factor required for effective nodulation [128] and involved in the differential expression of over 200 genes, including many that regulate redox homeostasis [129]. Deletion of *lsrB* resulted in poor growth and increased sensitivity to the detergent deoxycholate [127], and *lsrB* orthologs in *B. abortus* and *A. tumefaciens* contribute to pathogenesis [130–133]. In addition, a significant fraction of the JspA transcriptome consists of genes of unknown function, and changes in their expression may promote fitness during host colonization as well. Many uncharacterized genes in the ExoS-ChvI regulon, and by extension the JspA transcriptome, are predicted to be translocated out of the cytoplasm and

envelope-associated, making them more likely to interact with the host environment and to maintain barrier integrity [30].

In particular, *jspA* was shown to confer resistance to the nodule-specific antimicrobial peptide NCR247 [63]. One possible explanation is that JspA changes gene expression patterns via ExoS-ChvI to counter such host defenses. Nevertheless, other possible explanations, not mutually exclusive, can also account for JspA's involvement in resistance against cell envelope assaults: for example, JspA may degrade other substrates or signaling pathways under specific conditions, or JspA and LppA may assist in the proper construction of the cell envelope by ensuring proper maturation of other lipoproteins. Furthermore, a number of other signaling systems, such as ActK-ActJ, CenK-CenR, CpxA-CpxR, EmmB-EmmC, FeuQ-FeuP, and NtrX-NtrY, contribute to the maintenance of cell envelope integrity in *S. meliloti* [46,55,134–143], and how these different systems cooperate with ExoS-ChvI to ensure successful symbiosis remains ripe for further investigation.

## Materials and methods

### Bacterial strains, growth conditions, and genetic manipulations

All *Sinorhizobium meliloti* strains used in this study are derived from Rm1021 [144], and all *S. medicae* strains are derived from WSM419 [145]; they are listed in S11 Table. Other alpha-proteobacterial strains used were *S. fredii* NGR234 [146,147] and *C. crescentus* NA1000 [148]. *E. coli* strains DH5α and DH10B (both from Invitrogen) were used for molecular cloning, gene expression, and maintenance of plasmids, which are listed in S12 Table. *Sinorhizobium* strains were cultured at 30°C in LB, TY, or PYE media; *C. crescentus* was cultured at 30°C in PYE; and *E. coli* strains were cultured at 30 or 37°C in LB [44]. When appropriate, antibiotics, agar, sucrose, and/or calcofluor were added at previously published concentrations [44,149]. IPTG and taurine were added as inducers [83] at concentrations described in the text. For pH shifts, LB medium was buffered with 20 mM 2-(*N*-morpholino)ethanesulfonic acid (MES), 3-(*N*-morpholino)propanesulfonic acid (MOPS), or Tris, and adjusted to pH 6, 7, or 8.5, respectively, with HCl or NaOH. Growth of cultures was monitored by measuring absorbance at 600 nm ($A_{600}$), with aliquots from tubes or flasks or with a BioTek Synergy H1m plate reader if grown in 48-well plates. N3-mediated generalized transduction, mobilization of plasmids from *E. coli* to *Sinorhizobium* or *C. crescentus* strains via triparental mating, and two-step allelic replacement by homologous recombination were all performed as previously described [44,150–154]. Standard techniques were used for manipulation and analysis of DNA, including PCR amplification, restriction digests, agarose gel electrophoresis, ligation, and transformation [155,156]. Plasmids and DNA fragments were purified using commercial kits (Qiagen). Elim Biopharmaceuticals synthesized custom oligonucleotides and provided Sanger DNA sequencing services.

### Expression in *E. coli*

ExoR-FLAG was expressed in *E. coli* from an IPTG-inducible $P_{lac}$ promoter on pMB859, derived from the pSRKKm vector [95]. To co-express JspA-HA, JspA$_{E148A}$-HA, and LppA-HA, we constructed plasmids derived from pDSW204, which is compatible with pSRKKm and also allows IPTG induction with a weakened $P_{trc}$ promoter [102]: pJC720, pJC730, pJC731, pJC733, pJC734, pJC735, pJC736, and pJC737 (S7 Fig and S1 File). Plasmids pJC720 and pJC733 carry wild-type *jspA-HA* and mutant *jspA$_{E148A}$-HA*, respectively, including 18 nucleotides upstream of *jspA*'s annotated start codon. We added *lppA-HA* to these plasmids in three different configurations and assessed their expression empirically. For pJC730 and pJC735, the ribosome binding site (RBS) of *E. coli araB* [157] was appended upstream of

*lppA-HA* and inserted after *jspA-HA* or *jspA$_{E148A}$-HA*. For pJC731 and pJC736, *lppA-HA*, along with 54 nucleotides upstream of its originally annotated start codon (18 nucleotides upstream of the new start codon suggested in this report), was inserted after wild-type or mutant *jspA-HA*. For pJC734 and pJC737, *lppA-HA* and its upstream sequence were inserted in front of wild-type or mutant *jspA-HA*. We intended the RBS of *araB* to enhance expression of *lppA-HA* in *E. coli*, but that configuration (pJC730 and pJC735) yielded the lowest levels of expression (S7 Fig). Because *lppA-HA* is in-frame with wild-type or mutant *jspA-HA* on pJC731, pJC734, pJC736, and pJC737, read-through translation [158] appeared to produce low levels of fusion proteins that were sometimes detectable on Western blots (S7 Fig).

## Homology and domain analysis

Orthologs in representative genomes and their sequence similarities to the query were determined via BLAST [80]. Genomic contexts are presented as annotated in the National Center for Biotechnology Information (NCBI) database [159]. Protein domains were predicted using InterPro [160], Pfam [161], and LipoP [81].

## Calcofluor assays

EPS-I production was assessed as previously described [44], with LB plates containing 0.02% calcofluor white M2R (MP Biomedicals). Liquid cultures were calibrated to the same optical density (A$_{600}$ of 0.2–0.5) and serially diluted ten-fold in water, and four or five µL of the $10^{-2}$ to $10^{-6}$ dilutions were each spotted onto calcofluor plates containing appropriate additives, such as taurine for induction and oxytetracycline for plasmid selection. Dilutions were at times spotted onto PYE plates as well for comparison. Plates were examined and photographed after 3–4 days of incubation with a Kodak 4000MM Pro Image Station, with its associated Carestream MI software and filters (430 nm excitation and 535 nm emission). The fluorescence intensity of each spot was standardized relative to the corresponding wild-type control on the same plate, and the average values of the $10^{-2}$ to $10^{-4}$ spots from at least three independent plates were compared.

## β-glucuronidase (GUS) assays

Transcriptional fusions for β-glucuronidase (GUS) assays were constructed, and GUS activities in different strains under various growth conditions were determined, as previously described [44,162]. Fusions to *uidA* were introduced into the genome using nonreplicating plasmids, and the wild-type function of the corresponding gene was preserved (S12 Table, S1 File). Cells were lysed after measuring the optical density of the culture (A$_{600}$), and PNPG (*p*-nitrophenyl-*β*-D-glucuronide) was incubated with the lysed cells until the mixture turned light yellow, when absorbance at 415 nm (A$_{415}$) was measured. GUS activity was derived according to the formula: A$_{415}$ x 1000 / [(incubation time in minutes) x (culture volume in mL) x A$_{600}$]. *p* values and statistical significance were determined using t-test (two-tailed, unequal variance).

## Symbiosis assays

Symbiotic association between *Sinorhizobium* strains and *Medicago* plants was assessed as previously described [44,137,163,164]. Alfalfa (*M. sativa* GT13Rplus) and barrel medic (*M. truncatula* cultivar Jemalong) were cultivated individually in 18x150-mm glass tubes on agar slants made with standard nodulation medium (as described in [165], except with 2 mM KH$_2$PO$_4$ and 0.5 mM MES, pH 6.3) and 11.5 g/L Phyto agar (PlantMedia); seeds were surface-sterilized with 70% ethanol and 50% bleach, rinsed with water, germinated in inverted 100x25-mm Petri

dishes, placed on agar slants, and allowed to grow for three days at 22°C under fluorescent lamps (16-h day length) before inoculation. *M. sativa* was inoculated with *S. meliloti* Rm1021 and its derivatives, while *M. truncatula* was inoculated with *S. medicae* WSM419 and its derivatives because WSM419 is a better symbiotic partner for *M. truncatula* compared to *S. meliloti* strains, including Rm1021 [85–87]. Bacterial cells grown to mid-logarithmic phase were suspended in water to an $A_{600}$ of 0.1, and each seedling was inoculated with 0.1 mL of the suspension [approximately $10^7$ colony-forming units (CFU)]. The numbers of white and pink nodules that developed on plant roots were recorded at 14, 21, and 28 days post-inoculation (dpi). Nodules are initially white and then turn pink due to production of leghemoglobin, indicative of nitrogen fixation [7]. For competitive colonization assays, equal volumes of two cell suspensions (with $A_{600}$ of 0.1) were mixed and then diluted ten-fold, and each seedling was inoculated with 0.1 mL of the diluted mixture (approximately $10^6$ CFU). (For three of the 12 competitive assays with *S. medicae* strains, trials D1, D3, and E1, the inoculating mixtures were not diluted, and each seedling received $10^7$ CFU). The CFU and ratios of strains in the inoculating mixtures were determined by plating serial dilutions on PYE containing streptomycin, nalidixic acid, neomycin, or spectinomycin. Symbiosis competitiveness was assessed 28 dpi by harvesting nodules, surface-sterilizing them individually with 10% bleach, crushing each in PYE medium, and plating serial dilutions of the extracts: 10 μL each of $10^{-1}$ to $10^{-4}$ dilutions were dripped to form lines on plates, and colonies were counted after three to four days of incubation at 30°C. A nodule was considered to be dominated by a particular strain if more than 80% of the CFU from the extract can be attributed to that strain. Consistent with previous reports [44,88], the majority of nodules were dominated by a single strain (Fig 4 and S4 Table). For some nodules not dominated by a single strain, colonies recovered on permissive plates (containing streptomycin for Rm1021 and its derivatives, or nalidixic acid for WSM419 and its derivatives) were re-streaked to verify the ratios of strains (S4 Table, "patch" columns). In those cases, the nodule was assigned as mixed occupancy if neither strain gave rise to at least 90% of the colonies tested. In some instances, two or more adjacent nodules were harvested together and crushed in the same tube. If such a sample was dominated by a single strain, then it counted as a single nodule for that strain. On the other hand, if the sample yielded a mixture of two strains, then it was excluded from the final tally of that particular competition trial [S4 Table, "Mixed (Multi. Nodules)"]. In Fig 4, *p*-values were calculated using the Mann-Whitney-Wilcoxon test (two-tailed) for the *M. sativa* competitions and the t-test (two-tailed, unequal variances) for the *M. truncatula* competitions. The Mann-Whitney-Wilcoxon test is nonparametric but has less power for smaller sample sizes (and is ineffective for a total sample size less than eight) [166]; thus, the t-test was more suitable for analyzing the *M. truncatula* competitions, which had fewer trials per category compared to the *M. sativa* competitions. S4 Table provides the t-test *p*-values for competitions in both host plants.

## Transcriptome analysis

Microarray analysis of RNA transcripts using custom Affymetrix GeneChips was conducted as previously described [43,92]. RNA purification, cDNA preparation, chip hybridization, fluidics, scanning, and data analysis were performed accordingly [92]. Strains JOE3200 (carrying pCM130 vector), JOE4140 (expressing wild-type *jspA*), and JOE4400 (expressing mutant *jspA*$_{E148A}$) were grown in PYE supplemented with 0.5 μg/mL oxytetracycline and 10 mM taurine for 4.5 hours to mid-logarithmic phase, when cells were harvested for RNA extraction. Three biological replicates were used for each strain analyzed. Nine pairwise comparisons were made between two strains: a change in signal was considered significant if $p \leq 0.05$. As with other gene array platforms, our DNA chip measures mRNA abundance, which is

influenced by both transcription and mRNA decay; we use the term "expression" to include the sum of all factors affecting mRNA abundance. Raw microarray data have been deposited in the NCBI Gene Expression Omnibus [167] under Accession GSE155833. The significance of overlap between transcriptomes or sets of genes was determined using hypergeometric probability test, as previously described [30,98]. Genes in the ExoR/ExoS transcriptome were deduced from interrogation of *exoR95*::Tn5 and *exoS96*::Tn5 mutants that have hyperactive ChvI activity [21]; thus, many of the changes in gene expression may be indirect or independent of the ExoS-ChvI pathway. In addition, two biological replicates for each strain were used in that experiments (instead of the three used in the current analysis for JspA transcriptome), and that could have contributed to the relatively large number of genes in the ExoR/ExoS transcriptome. The ChvI regulon contains a combination of nonredundant genes from two publications: (1) those that decreased expression in the ChvI(K214T) partial-loss-of-function mutant and increased expression in the ChvI(D52E) gain-of-function mutant compared to wild type [36], and (2) group 1 and group 2 direct targets of ChvI [30].

## Western blotting

Immunoblotting was performed using standard procedures [155,156]: 1.5 mL of culture samples were collected, resuspended in SDS sample buffer (at 150 μL for culture $A_{600}$ of 1), boiled for 5 min, resolved by SDS-PAGE, and transferred to PVDF membrane for detection by chemiluminescence (SuperSignal West Pico). Monoclonal anti-V5 (Invitrogen R960-25; AB_2556564) was used at 1:2500 dilution, anti-HA [clone 2–2.2.14] (Thermo 26183; AB_10978021) used at 1:1000 to 1:5000 dilution, and antibody to *E. coli* ATPase B [7E3F2] (Abcam ab110280) used at 1 ng/mL. Peroxidase-conjugated monoclonal anti-FLAG M2 antibodies (Sigma A8592) were used at 1:2000 to 1:5000 dilution. Peroxidase-conjugated donkey anti-mouse IgG antibodies (715-035-150) were from Jackson ImmunoResearch Lab and used at 1:25,000 dilution. To examine blots containing both HA and FLAG epitopes, we probed the blots first with anti-FLAG antibodies, detected bound antibodies with chemiluminescent substrates, washed the blots with Tris-buffered saline [155] containing 0.05% Tween 20 (TBST), then probed with anti-HA primary antibodies and peroxidase-conjugated secondary antibodies, and imaged again with chemiluminescence. Similarly, to detect both V5 epitope and the beta subunit of ATP synthase, we probed blots first with anti-V5 antibodies, imaged with chemiluminescence, washed the blots with TBST, then probed with antibodies against ATPase B, and treated with chemiluminescence reagents. Images captured in the second detection showed both target epitopes. Each blot image shown is representative of at least two biological replicates.

## Supporting information

**S1 Fig. Production of calcofluor-binding exopolysaccharides in *S. medicae* WSM419 and its derivatives.** Ten-fold serial dilutions of logarithmic-phase cultures were spotted onto solid media and allowed to grow for three days prior to imaging. (**A**) Representative images show fluorescence of wild-type WSM419, Δ*jspA* (ΔSmed_3110) mutant, Δ*lppA* (ΔSmed_0632) mutant, and derivatives marked with neomycin (Nm^R) or spectinomycin (Sp^R) resistance [*nptII* or *aadA* linked to *podJ* (Smed_0147) or replacing *jspA*] on LB plates containing calcofluor. Darker spots indicate brighter fluorescence. (**B**) WSM419 and Δ*jspA* and Δ*lppA* mutants carrying the vector (pCM130) or a plasmid with *S. meliloti jspA*, *jspA*_E148A, or *lppA* under the control of a taurine-inducible promoter (pJC535, pJC555, or pJC532, respectively) were grown on LB plates containing tetracycline (Tet) and calcofluor, without or with taurine (5 mM taurine for *jspA* complementation, 10 mM for *lppA*). Visible-light images of corresponding strains

grown on PYE plates show mucoid colonies. Labels on the left indicate strain numbers, while labels on the right indicate genotypes. Each experiment was performed at least twice.
(PDF)

**S2 Fig. Expression of mutant *lppA* and *jspA* alleles in *S. meliloti*.** (**A**) Calcofluor fluorescence was used to assess EPS-I production in Δ*lppA* mutants expressing different alleles of *lppA* from a taurine-inducible promoter. (**B**) Residues altered in the LppA variants are depicted as ball-and-stick representations in the AlphaFold structural prediction (UniProt Q92R89) (which does not contain the additional 12 residues at the N-terminus that optimize the signal sequence). (**C**) Wild-type Rm1021 expressing different *jspA* alleles from a taurine-inducible promoter exhibit varying levels of fluorescence on calcofluor plates. (**D**) Immunoblot shows steady-state levels of different versions of JspA-HA. E148A-HA, E148D-HA, and H147A-HA stand for mutant versions of JspA-HA, encoded by *jspA*$_{E148A}$-*HA*, *jspA*$_{E148D}$-*HA*, and *jspA*$_{H147A}$-*HA*, respectively. Samples were harvested from wild-type strains grown in LB with or without 10 mM taurine (+ or - taurine) for 3 hours. Numbers to the right of immunoblot indicate approximate molecular mass standards, in kDa. Plasmids pJC532, pJC605, pJC606, pJC607, pJC608, and pJC609 were used for expressing *lppA*, *lppA*$_{C23S}$, *lppA*-*HA*, *lppA*$_{C23S}$-*HA*, *lppA*$_{G96W}$-*HA*, and *lppA*$_{A78S}$-*HA*, while pJC535, pJC555, pJC556, pJC557, pJC558, pJC559, pJC560, pJC561 were used for *jspA*, *jspA*$_{E148A}$, *jspA*$_{E148D}$, *jspA*$_{H147A}$, *jspA*-*HA*, *jspA*$_{E148A}$-*HA*, *jspA*$_{E148D}$-*HA*, and *jspA*$_{H147A}$-*HA*, respectively. Vectors used were pCM130 (A, D) or pJC478 (C). For assessing calcofluor fluorescence, ten-fold serial dilutions ($10^{-2}$ to $10^{-5}$) of logarithmic-phase cultures were spotted onto LB plates without or with taurine, and allowed to grow for three days prior to fluorescence imaging. Darker spots on representative images indicate brighter fluorescence. Portions of panels (A) and (C) are the same as images shown in Fig 6A and 6C.
(PDF)

**S3 Fig. Overexpression of *jspA* alleles in *S. medicae* WSM419, *S. fredii* NGR234, and *C. crescentus* NA1000.** Ten-fold serial dilutions of logarithmic-phase cultures were spotted onto PYE plates containing 0, 5, or 10 mM taurine. *C. crescentus* NA1000 derivatives were grown with 1 μg/mL oxytetracycline for two days, while *Sinorhizobium* WSM419 and NGR234 derivatives were grown with 5 μg/mL oxytetracycline for three days at 30°C prior to imaging. Labels on the left indicate strain numbers, while labels on the right indicate the *jspA* alleles being expressed from a plasmid. Plasmids used were pJC614 (*jspA*), pJC615 (*jspA*$_{E148A}$), pJC616 (*jspA*-*HA*), and pJC617 (*jspA*$_{E148A}$-*HA*). Images shown represent four replicates on two different days.
(PDF)

**S4 Fig. Depletion of ChvI.** (**A**) Plate images show growth of ChvI depletion strain on LB medium in the presence and absence of 1 mM IPTG. The top strain (AD124) carried the pSRKKm vector and had a Δ*chvI* allelic replacement plasmid (pAD112) integrated into its chromosome but retained a copy of *chvI*⁺, while the bottom strain (AD115) carried pAD101, with *chvI* under the control of the P$_{lac}$ promoter, and had its chromosomal *chvI* replaced by a hygromycin resistance gene (*hph*). (**B**) Plots show growth curves of ChvI depletion strains over 24 hours in LB, in the presence or absence of 0.5 mM IPTG. Strains with chromosomal *chvI*⁺ or Δ*chvI* alleles carried pAD101, as well as a compatible vector (pCM130) or derivatives (pJC535 or pJC555) containing taurine-regulated *jspA* or *jspA*$_{E184A}$ (abbreviated as E148A), under control of the P$_{tau}$ promoter. No taurine was added in these growth experiments. Cultures were shaken at 1000 rpm in 48-well plates, with 0.4 mL medium (containing kanamycin and oxytetracycline) per well. Absorbance at 600 nm (A600) was measured every 30 minutes.

Average readings for three different days are shown, with corresponding color shadings indicating standard deviations. In the presence of IPTG, all strains exhibited similar growth patterns; curves for depletion strains carrying P$_{tau}$-*jspA* or *jspA*$_{E148A}$ grown with IPTG were omitted for clarity. Strains shown here for growth curves (JOE5579, JOE5604, JOE5606, JOE5608) all contain a genomic *exoY-uidA* reporter and constitute a subset of those used for GUS assays in Fig 8B. Absorbance readings and generation times calculated from growth curves are available in S9 Table.
(PDF)

**S5 Fig. Growth curves of *exoR-V5* strains expressing *jspA* alleles.** (**A**) Strains JOE5242 (carrying the pSRKGm vector), JOE5244 (with *jspA* on pJC652), and JOE5246 (with *jspA*$_{E148A}$, noted as E148A, on pJC653) were grown in 48-well plates, with 0.4 mL PYE per well, in the presence of absence of 1 mM IPTG. Absorbance at 600 nm (A600) was measured every 30 minutes. Average readings for three different days are depicted, with surrounding shadings indicating standard deviations. Lines without markers represent growth in the absence of IPTG (-); standard deviations for these were omitted for simplicity. Fig 7 shows a portion of this graph. (**B, C**) Liquid cultures of the same strains were grown in flasks and induced with 1 mM IPTG in (B) PYE or (C) LB medium, and A600 was measured every hour for 12 hours. The plots for (B) and (C) were generated from single experiments. Absorbance readings and generation times calculated from growth curves are available in S9 Table.
(PDF)

**S6 Fig. Steady-state levels of ExoR-FLAG when co-expressed with LppA and different forms of JspA in *C. crescentus* NA1000.** JspA, LppA, and ExoR-FLAG expression is indicated above the blot: - signifies no expression, + signifies expression of the wild-type allele, and \*, #, and ^ represent JspA$_{E148A}$, JspA-HA, and JspA$_{E148A}$-HA, respectively. Positions of bands representing JspA-HA and ExoR-FLAG are indicated to the right of the blot; "pre" indicates the precursor form of ExoR-FLAG, "ExoR-FLAG" the mature form, and "deg" a major degradation product. Approximate molecular mass, in kDa, are shown to the left of the blots, while lane numbers are shown below. Expression of ExoR-FLAG from pMB859 was induced with 0.1 mM IPTG in PYE medium for 4 hours, while no expression means that the strain carried the empty vector pSRKKm under the same growth conditions. Expression of LppA and different versions of JspA were induced with 10 mM taurine from the following plasmids: lanes 1 and 4, empty vector pJC473 when neither expressed (- for both LppA and JspA); lanes 2 and 9, pJC616 for JspA-HA only (red #); lanes 3 and 10, pJC617 for JspA$_{E148A}$-HA (red ^); lane 5, pJC614 for wild-type JspA (black +); lane 6, pJC615 for JspA$_{E148A}$ (black \*); lane 7, pJC702 for LppA and JspA (black + for both); lane 8, pJC706 for LppA and JspA$_{E148A}$ (black + and \*); lane 11, pJC707 for LppA and JspA-HA (black + and red #); and lane12, pJC708 for LppA and JspA$_{E148A}$-HA (black + and red ^). The blot was first probed with anti-FLAG antibodies and then with anti-HA antibodies. This representative image was captured after both antibodies were applied.
(PDF)

**S7 Fig. Steady-state levels of ExoR-FLAG when co-expressed with wild-type or mutant JspA-HA and varying levels of LppA-HA in *E. coli* DH10B.** Symbols in rows above the blots indicate expression of different proteins, with - indicating no expression. For ExoR-FLAG, + signifies expression from pMB859, while - signifies carriage of the pSRKKm vector. For JspA-HA, red # and ^ indicate expression of wild-type JspA-HA and mutant JspA$_{E148A}$-HA, respectively. For LppA-HA, the size of the + symbol reflects the level of expression, with the orange, bold + indicating the highest levels. Schematics above the blots represent gene

arrangements on plasmids used for expressing JspA-HA variants and LppA-HA: red ^ indicates plasmids that carry the $jspA_{E148A}$-HA mutant allele and the approximate location of the active site mutation in the gene; RBS preceding *lppA-HA* in pJC730 and pJC735 is the ribosome binding site of *E. coli araB*. Positions of bands representing JspA-HA, LppA-HA, and ExoR-FLAG are indicated to the right of the blots: "pre" indicates the precursor form of ExoR-FLAG, "ExoR-FLAG" the mature form, and "deg" a major degradation product; "readthrough" indicates presumed fusions of JspA-HA and LppA-HA that resulted from translational read-through. LppA-HA expression is shown with two different blots because signals from strains with lower levels of expression (lanes 4, 5, 6, 7) were overwhelmed by those with the highest levels of expression (lanes 8, 9, 12, 13); the bottom image was captured using a duplicate blot without the highest-expression samples. Steady-state levels of pre-processed ExoR-FLAG appeared relatively high when expressed in *E. coli* DH10B compared to *S. meliloti* and *C. crescentus*, possibly due to inefficient export. Approximate molecular mass, in kDa, are shown to the left of the blots, while lane numbers are shown below. All strains were induced with 0.1 mM IPTG in LB medium for 4 hours. Plasmids used for expressing JspA-HA and LppA-HA are indicated in the schematics above the blots; lanes 1 and 14 used pDSW208 (expressing GFP) as the vector, lanes 10 and 11 used pJC720 and pJC733, while lanes 12 and 13 used pJC734 and pJC737, respectively.
(PDF)

**S1 Table. Orthologs of JspA, LppA, and ExoR in representative bacterial species.**
(XLSX)

**S2 Table. Calcofluor fluorescence of strains.**
(XLSX)

**S3 Table. Number of pink and white nodules on individual *M. sativa* or *M. truncatula* seedlings inoculated with *S. meliloti* or *S. medicae* strains.**
(XLSX)

**S4 Table. Compilations of symbiosis competitions between *S. meliloti* strains in *M. sativa* or between *S. medicae* strains in *M. truncatula*.**
(XLSX)

**S5 Table. Expression of GUS fusion reporters.**
(XLSX)

**S6 Table. JspA-regulated genes.**
(XLSX)

**S7 Table. Hypergeometric probability tests for overlaps between JspA, ExoRS, and RpoH1 transcriptomes and ChvI regulon.**
(XLSX)

**S8 Table. Hypergeometric probability tests for overlap of JspA transcriptome with genes that altered expression in NCR247-treated cells or the ΔpodJ1 mutant.**
(XLSX)

**S9 Table. Growth curve data for ChvI depletion and JspA overexpression.**
(XLSX)

**S10 Table. Overlap of JspA transcriptome with genes that changed expression upon acid stress.**
(XLSX)

**S11 Table.** *Sinorhizobium meliloti* **and** *medicae* **strains used in this study.**
(DOCX)

**S12 Table. Plasmids used in this study.**
(DOCX)

**S1 File. Plasmid sequences.**
(ZIP)

## Acknowledgments

We thank Esther Chen, Mike Kahn, and Bob Fisher for discussion, support, and carrot cake. Esther Chen also provided critical feedback on the manuscript. We are grateful to past and current laboratory members for their assistance in various aspects of this project, particularly Reyna Benitez, Tanisha Saini, and Noel Lejat for performing GUS assays.

## Author Contributions

**Conceptualization:** Sharon R. Long, Joseph C. Chen.

**Data curation:** Melanie J. Barnett, Joseph C. Chen.

**Formal analysis:** Melanie J. Barnett, Joseph C. Chen.

**Funding acquisition:** Sharon R. Long, Joseph C. Chen.

**Investigation:** Julian A. Bustamante, Josue S. Ceron, Ivan Thomas Gao, Hector A. Ramirez, Milo V. Aviles, Demsin Bet Adam, Jason R. Brice, Rodrigo A. Cuellar, Eva Dockery, Miguel Karlo Jabagat, Donna Grace Karp, Joseph Kin-On Lau, Suling Li, Raymondo Lopez-Magaña, Rebecca R. Moore, Bethany Kristi R. Morin, Juliana Nzongo, Yasha Rezaeihaghighi, Joseph Sapienza-Martinez, Tuyet Thi Kim Tran, Zhenzhong Huang, Aaron J. Duthoy, Melanie J. Barnett, Joseph C. Chen.

**Project administration:** Sharon R. Long, Joseph C. Chen.

**Supervision:** Sharon R. Long, Joseph C. Chen.

**Visualization:** Melanie J. Barnett, Joseph C. Chen.

**Writing – original draft:** Joseph C. Chen.

**Writing – review & editing:** Aaron J. Duthoy, Melanie J. Barnett, Sharon R. Long, Joseph C. Chen.

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
