## [Decision Letter · Decision Letter 0]

27 Jun 2023

Dear Dr Chen,

Thank you very much for submitting your Research Article entitled 'A protease and a lipoprotein jointly modulate the conserved ExoR-ExoS-ChvI signaling pathway critical in Sinorhizobium meliloti for symbiosis with legume hosts' to PLOS Genetics.

The manuscript was fully evaluated at the editorial level and by independent peer reviewers. The reviewers appreciated the attention to an important problem, but raised some concerns about the current manuscript. Based on the reviews, we cannot accept this version of the manuscript, but we would be willing to review a much-revised version. We cannot, of course, promise publication at that time.

If you decide to revise the manuscript for further consideration at PLOS Genetics, please aim to resubmit within the next 60 days, unless it will take extra time to address the concerns of the reviewers, in which case we would appreciate an expected resubmission date by email to plosgenetics@plos.org.

We hope that you will consider submitting a revision of your manuscript. Please do not hesitate to contact us if you have any concerns or questions.

Yours sincerely,

Aimee Shen

Academic Editor

PLOS Genetics

Lotte Søgaard-Andersen

Section Editor

PLOS Genetics

The Reviewers recognized the importance of the authors’ findings in delineating a novel mechanism for controlling the activation of a conserved two-component signaling pathway in the Alphaproteobacteria that regulates interactions with eukaryotic hosts. They were particularly enthusiastic regarding the the authors’ work in contributing to a broader understanding of the role that lipoproteins play in signaling. However, they noted a number of areas where the manuscript could be improved to merit publication. Please address their comments with particular attention to streamlining the presentation of the data and Introduction, refining their model regarding the role of LppA and JspA in signaling through the ExoR-ChvI-ChvG pathway (Reviewer 2), and testing the membrane localization of LppA and JspA directly. The rigor of the data presented would be improved by the addition of load controls to Figures 9 and 10, and Figure 2 is missing statistical analyses. Please also standardize the data presentation to show dots-on-plots.

Reviewer's Responses to Questions

**Comments to the Authors:**

Reviewer #1: Summary:

The manuscript from Bustamante et al. reports analysis of two presumptive lipoproteins LppA and JspA in Sinorhizobium meliloti (Sm) identified based on suppression of cell envelope defects in a Sm podJ mutant. These suppressor mutations also lead to decreased production of the EPS-1 polysaccharide. The JspA sequence identifies it as a periplasmic metalloprotease and the lipoprotein LppA is found to be required for its activity. Starting with the EPS phenotype, extensive genetic and expression analyses reveal that JspA and LppA act through the ExoR-ExoS-ChvI pathway that regulates EPS-1 synthesis in Sm and other rhizobia. Mutants manifest a competitive defect relative to wild type in nodulation assays on two different legumes. Mutants in jspA lead to elevated steady state ExoR levels, and overexpression leads to lower levels, and the presence of LppA is required for ExoR destabilization. Expression of these Sm proteins in Caulobacter crescentus and E. coli leads to a largely similar effect on ExoR turnover.

Major Comments:

This is a well-performed study that does a nice job of defining the impact of LppA and JspA and the way they act through ExoR-ExoS-ChvI. The conservation of these proteins suggests that they may have similar roles in other Alphaproteobacteria. The strength of the effects observed is surprisingly modest but statistically significant and consistent at almost all levels in the different analyses utilized. The role of periplasmic proteolysis is particularly relevant for ExoR-dependent regulation as low pH stimulates cleavage of ExoR, leading to high levels ExoS-ChvI activity. The paper is written somewhat opaquely about the relationship of the JspA-LppA system with the protease responsible for the low pH activation of ExoS-ChvI through ExoR cleavage. JspA does not appear to be responsible for the observed regulatory cleavage of ExoR, but it does seem to control the overall levels of the protein in the periplasm, which clearly can impact the regulatory patterns observed. It seems plausible that JspA and LppA play more general roles in the periplasm, and that ExoR is one of their substrates. The authors seem to lean heavily towards a specific role with the ExoR pathway, but remain somewhat obtuse about the low pH control. The authors should be more direct in their interpretation of their findings in the context of the ExoR proteolysis model.

The manuscript presents an impressive amount of data and a wide range of experiments, but in some cases this seems to be inefficiently organized and presented. For example, the Calcofluor fluorescence plates could be presented as fluorescence values and the dilution plates could be supplemental information – do we really need to see all of these dilution plates? The Introduction also seems unnecessarily comprehensive.

Minor Comments:

1. Title – Both LppA and JspA are lipoproteins so this title is inaccurate.

2. L71. “Essential” is a loaded term – I would use something else – “critical” perhaps?

3. L230-1. Based on the sequences presented, it is predicted that lppA and jspA would localize to the outer membrane. Is there a reason to be so circumspect about this prediction? Although the role of lppA lipidation is genetically examined, that of jspA is not. Why not?

4. L247. It is debatable whether the Ptau-jspA expressed in the jspA mutant causes a decrease in colony formation at 5 mM taurine, although this is clear at 10 mM. In stating that this effect is not observed in a lppA mutant, only 5 mM taurine is tested and 10 mM is not shown. Why is this not included to make the comparison valid? In contrast, the comparison to wt with Ptau-jspA is convincing.

5. L294. Most of the Miller unit values provided in the Table are strikingly low in the range around 10 units. What are the limits of detection with the assays as performed? Is there a chance that the increases are greater than represented and perhaps the decreases also underrepresented

6. L304. If the plasmid-borne gene is induced with taurine why would it be described as constitutive?

7. L361. Should indicate this is C. crescentus here to avoid confusion.

8. L501. Not sure I agree this is modest. Were growth rates calculated? Is it interesting that the impact of the catalytic site mutant is different between the two growth formats?

9. L522-3. Surprising that the authors do not mention that this increase seems specific to the processed form, and less so or not at all with the pre-processed form, consistent with the periplasmic location for this degradation activity.

10. L531-3. Figure 9. One might argue that the lppA mutant is diminished for ExoR-FLAG in the strain with JspHA and chromosomal JspA (lane B5) compared to the strain with no JspA (lane B4) or even just the chromosomal copy (lane B6). Maybe there is ExoR degradation in the absence of LppA if sufficient JspA is expressed? Is it worth quantitating these ExoR bands?

11. L539. IPTG-inducible Plac promoter

12. L547-8. Its confusing to have both the untagged JspA and the Jsp-HA indicated in the same line for the gel key in 10A. I suggest adding a separate line for the two different expression constructs. The distinction between the black and red symbols is not particularly clear (especially for people, like this reviewer, with varying degrees of colorblindness).

13. L552. It is also striking that the pre-processed form of ExoR is so much stronger in DH10B. Is this likely due to less efficient recognition of the ExoR signal sequence? This should at least be acknowledged in the results, even if no experiments were performed to determine the reason for it.

14. L548-51. The 20 kDa species mentioned here should be marked or labeled in some way on 10A and S5A.

15. L595. In the case of Williams et al. this inhibition appeared to be ExoR-independent, in contrast to the LppA/JspA affect reported here.

16. L693. C. crescentus? Only this single species, correct, rather than multiple representatives of the genus as with Sinorhizobium.

17. L772. The sentence is confusing – is it that white nodules eventually turn pink or that at first they turn pink and then lose this pigmentation.

18. L995. Presence OR absence. Also, is it clear that the leaky expression of chvI is depleted relative to the wild type?

19. L1049-50. Wild-type (+) or the jspAE148A mutant.

20. L1061. I assume this was probed with a mixture of anti-FLAG and anti-HA antibodies as this appears to be a single blot and exposure, rather than two separate probings. This is in contrast to 10B in that respect.

21. Tables 1-3. These tables are awkwardly formatted in a way that makes comparisons cumbersome – there is too much text to navigate through and they take up too much space. For example, Table 1 lists out the same mutant designations three times – why not list these 1 time and array the data from the three different gene horizontally?

22. Figure 10B – the asterisk for the band ascribed to the readthrough product is not mentioned in the results and is rather buried at the end of a long figure legend.

Reviewer #2: In this paper, Bustamente and colleagues report that two novel lipoproteins, LppA and JspA, are regulators of the key two-component signaling pathway ExoS-ChvI in Sinorhizobium meliloti. The authors use deletion mutants and genetic analysis to determine that LppA and JspA work in the same pathway to positively regulate exopolysaccharide (EPS) production. These factors are likely to be sensors of stressors/signals and appear to trigger response by promoting the degradation of ExoR, a negative regulator of the system. Ultimately, these proteins promote EPS production and regulate other factors required for infection through the ExoR-ExoS-ChvI system. The importance of these proteins for host infection is shown by the competitive defect of lppA and jspA single mutants relative to WT strains, although no clear defect is present in monoculture infections of the mutants.

Overall, the study is well-conducted, and sufficient controls are included to support the authors conclusions. The discovery of two novel lipoproteins regulating the activation of a two-component system contributes to a growing understanding of the role that lipoproteins play in signaling across the Gram-negative envelope. The study could be strengthened by a few changes to the writing and experiments to confirm/support interpretations made by the authors, which are listed here below as major and minor points.

Major points

1. The authors do not comment on where LppA and JspA are precisely localized (Lines 230-232). Ideally, confirming whether these proteins are localized to the IM or OM would provide more definitive idea as to the signals sensed by these proteins. This would also strengthen the comparisons the authors make to other lipoprotein envelope stress sensors, such as RcsF, which appear to be predominantly OM localized. Because the authors already have vectors expressing tagged versions of these proteins, conducting a localization experiment, such as sucrose density gradient fractionation, may be useful to include. Alternatively, it may be worth seeing if the introduction of a “canonical” Lol pathway avoidance signal to these proteins leads to different phenotypes, which may imply localization in the OM or IM normally, although this may be complicated by differences between species in Lol pathway function.

2. If IM-OM fractionation is not feasible, it would be useful to at least confirm that LppA and JspA are membrane-anchored in whole membrane preparations. This would strengthen the interpretation that C23S mutation in LppA impacts membrane anchoring. At this point, it’s difficult to make meaningful interpretations about the impact of LppA acylation because the C23S variant appears to be highly unstable, and its lack of change in calcofluor fluorescence could be attributable simply to low levels of protein present. It would be useful to create a periplasm-localized but stable version of LppA (e.g. by fusing mature LppA sequence to the signal peptide of a known periplasmic and/or IM-anchored protein). This would allow the authors to more definitively determine if the lipoprotein-nature/membrane anchoring of LppA is essential to its function versus the data provided by the unstable C23S mutant.

3. Similarly, I think it would be important that the authors test whether mutating the putative N-terminal cysteine of JspA has similar impacts on JspA function as it does to LppA. If membrane-anchoring is important for LppA, it would also make sense that JspA must also be membrane anchored but this is not tested in this study.

4. Are there loading controls for the protein expression blots (e.g. Figure 9 and 10)? While the conclusions reached in the text appear to be broadly accurate, the presence of loading controls (e.g. presumably the inclusion of anti-ATPase B) would allow the authors to quantify protein expression levels and make more quantitative claims about the relative levels of ExoR present. Also, I’m not sure that the C. crescentus and E. coli heterologous expression experiments shown in Figure 10 contribute enough novel information to be included as a separate main text figure. I think that they could be combined into Figure 9 or included as a supplemental figure with no loss to the quality of the claims made in the text.

Minor points

1. To better translate the authors’ mutant analysis to real impacts on infection, it would be useful test if the single amino acid substitutions (e.g. C23S in LppA and the enzymatically inactive E148A JspA) phenocopy their whole gene deletions in the plant infection model. Testing this could strongly support the in vivo mechanisms of these proteins in S. meliloti infection.

2. Lines 250-253: The way the data is presented is slightly confusing because the panels referred to in these lines are included in panels A and B, which were referred to earlier in the paragraph. It might be useful to take the bottom panels of A and B and combine them into a new panel that contains only the experiments where LppA or JspA are overexpressed in the opposing deletion mutant background.

3. Lines 253-255: The connection between lack of growth inhibition seen when overexpressing JspA in an lppA mutant and overall conclusion (that “LppA and JspA act in concert …”) could use a bit more explanation.

4. Line 320: The phrasing “LppA acts as a lipoprotein” might be somewhat confusing as lipoproteins are functionally diverse (as seen in the fact that JspA, a lipoprotein, is a protease). It might be useful to rephrase this to be more specific (e.g. “to determine if LppA membrane anchoring is important” etc.).

5. Line 341-342: While it’s very interesting that the mutants of LppA (G96W and A78S) possess phenotypes that differ from WT, I’m not sure if including these mutants contribute very much to the manuscript unless there is more work done as to their potential mechanism, especially considering that these mutants do not completely abrogate LspA function. It may be worth looking into the structure of LppA as predicted by AlphaFold (e.g. on the online AlphaFold database) to provide some rationale or hypothesis as to the potential effects of these mutations.

6. Line 440-441: It would be clearer if this sentence was rephrased to refer to the loss of JspA and LppA instead of directly referring to the proteins themselves.

7. Because of the comprehensive nature of this paper, it would be nice if the authors include a final model figure which incorporates all of their conclusions from their data to tie everything together.

Reviewer #3: I am extremely enthusiastic about this manuscript. It is an incredibly thorough genetic dissection of the the interaction with the predicted lipoprotein lppA and the predicted metalloprotease jspA with the exoR/exoS system and its regulation of the EPSI/flagella production switch. I would call it nearly flawless. On top of that, the manuscript does not stop with the genetics! It demonstrates that both JspA and LppA are required for degradation of ExoR, which is known to negatively regulate exoS/chvI sensor kinase system, which in turn, positively regulates expression of exoY.

The only change I recommend is a change in line 340 to state that 'these results are consistent with acylation at the Cys residue of the predicted lipobox allowing LppA...'

**Have all data underlying the figures and results presented in the manuscript been provided?**

Reviewer #1: **No: **Raw data for GUS fusion activity is not provided.

Reviewer #2: Yes

Reviewer #3: Yes

PLOS authors have the option to publish the peer review history of their article (what does this mean?). If published, this will include your full peer review and any attached files.

Reviewer #1: **Yes: **Clay Fuqua

Reviewer #2: No

Reviewer #3: **Yes: **Kathryn M. Jones

---

## [Decision Letter · Decision Letter 1]

2 Oct 2023

Dear Dr Chen,

Thank you very much for submitting your Research Article entitled 'A protease and a lipoprotein jointly modulate the conserved ExoR-ExoS-ChvI signaling pathway critical in Sinorhizobium meliloti for symbiosis with legume hosts' to PLOS Genetics.

The manuscript was fully evaluated at the editorial level and by independent peer reviewers. The reviewers appreciated the attention to an important topic but identified some concerns that we ask you address in a revised manuscript.

We therefore ask you to modify the manuscript according to the review recommendations. Your revisions should address the specific points made by each reviewer.

Yours sincerely,

Aimee Shen

Academic Editor

PLOS Genetics

Lotte Søgaard-Andersen

Section Editor

PLOS Genetics

We thank the authors for their careful attention to the comments of the Reviewers, and I look forward to accepting this excellent manuscript after the changes to the terminology regarding strains and mutants is modified based on Reviewer 1's comments.

Reviewer's Responses to Questions

**Comments to the Authors:**

Reviewer #1: Summary:

This extensively revised manuscript from Bustamante et al. describes studies on two presumptive lipoproteins LppA and JspA in Sinorhizobium meliloti (Sm) that they find impact the ExoR-ExoS-ChvI regulatory pathway through effects on ExoR stability in the periplasm.

Major Comments:

The authors have performed a commendable job of revising the manuscript in response to the initial review (particularly the comments of Reviewer 2 and my own comments). The manuscript includes a massive amount of data already, and I do agree with the authors in general that addition of extensive additional experimental data seems to go beyond reasonable expectations for a single paper. The study includes a wide range of experiments and approaches, and at least to me does a very impressive job of establishing the impact of lppA and jspA through the RSI pathway..

I did re-read the entire manuscript, and there was a minor issue that struck me on this re-reading that would be worth addressing. The authors use the term “strain” in a confusing way. WSM419 is a particular strain of S. medicae – a sub-lineage of the species. However, Sm1021 with an engineered deletion of jspA is most accurately described as a mutant of the strain 1021. The authors use these terms interchangeably, sometimes in the same sentence.

For example: - “While induction of lppA or 181 jspA expression promoted EPS-I production in wild type or corresponding deletion strains, lppA expression in the ΔjspA mutant (Fig 3A, bottom row) and jspA expression in the ΔlppA mutant (Fig 3B, bottom row) failed to increase calcofluor fluorescence (S2B Table). could clean up this terminology throughout the manuscript as it is confusing used.”

Are the strains different from the mutants? Given that the manuscript DOES in fact include multiple strains as well as a wide variety of different mutants, the usage of this terminology should be standardized.

Minor Comments:

L82. Ref 22 has nothing to do with the statement. Perhaps the wrong reference? Brucella is not a rhizobial endosymbiont so perhaps just using a more broad descriptor would fix this problem?

Reviewer #3: I did not have major criticisms of the first version. Recommendation is Accept

**Have all data underlying the figures and results presented in the manuscript been provided?**

Reviewer #1: Yes

Reviewer #3: Yes

PLOS authors have the option to publish the peer review history of their article (what does this mean?). If published, this will include your full peer review and any attached files.

Reviewer #1: No

Reviewer #3: **Yes: **Kathryn M. Jones

---

## [Editor Report · Decision Letter 2]

11 Oct 2023

Dear Dr Chen,

We are pleased to inform you that your manuscript entitled "A protease and a lipoprotein jointly modulate the conserved ExoR-ExoS-ChvI signaling pathway critical in Sinorhizobium meliloti for symbiosis with legume hosts" has been editorially accepted for publication in PLOS Genetics. Congratulations!

Yours sincerely,

Aimee Shen

Academic Editor

PLOS Genetics

Lotte Søgaard-Andersen

Section Editor

PLOS Genetics

Comments from the reviewers (if applicable):

**Data Deposition**

http://datadryad.org/submit?journalID=pgenetics&manu=PGENETICS-D-23-00529R2

**Press Queries**

---

## [Editor Report · Acceptance letter]

17 Oct 2023

PGENETICS-D-23-00529R2 

A protease and a lipoprotein jointly modulate the conserved ExoR-ExoS-ChvI signaling pathway critical in Sinorhizobium meliloti for symbiosis with legume hosts 

Dear Dr Chen, 

We are pleased to inform you that your manuscript entitled "A protease and a lipoprotein jointly modulate the conserved ExoR-ExoS-ChvI signaling pathway critical in Sinorhizobium meliloti for symbiosis with legume hosts" has been formally accepted for publication in PLOS Genetics! Your manuscript is now with our production department and you will be notified of the publication date in due course.

With kind regards,

Dorothy Lannert

PLOS Genetics

On behalf of:
